# Molecular landscape of the fungal plasma membrane and implications for antifungal action

Jennifer Jiang [1,2], Mikhail V. Keniya [3], Anusha Puri[1,2], Xueying Zhan [4], Jeff Cheng[1,2], Huan Wang [5], Gigi Lin[6], Yun-Kyung Lee[1,2], Nora Jaber[1,2,7], Caifeng Zhao[8], Cynthia Pang [1,2], Yasmine Hassoun[3], Haiyan Zheng [8], Erika Shor [3,9], Zheng Shi[5], Sang-Hyuk Lee[2,10], Min Xu [4], David S. Perlin [3] ✉ & Wei Dai [1,2] ✉

Fungal plasma membrane proteins represent key therapeutic targets for antifungal agents, yet their native structure and spatial distribution remain poorly characterized. Herein, we employ an integrative approach to investigate the organization of plasma membrane protein complexes in *Candida glabrata*, focusing on two abundant and essential membrane proteins, the β-(1,3)-glucan synthase (GS) and the proton pump Pma1. We show that treatment with caspofungin, an echinocandin antifungal that targets GS, disrupts the native distribution of membrane protein complexes and alters membrane biophysical properties. Perturbation of the sphingolipid biosynthesis further modulates drug susceptibility, revealing that the lipid environment plays an integral role in membrane protein organization and GS-echinocandin interactions. Our work highlights the importance of characterizing membrane proteins in their native context to understand their functions and inform the development of novel antifungal therapies.

Invasive fungal infections affect over a billion people and account for approximately 3.75 million deaths worldwide each year[1]. The most common opportunistic human fungal pathogens, particularly among individuals with pre-existing conditions, include *Candida* spp., *Aspergillus* spp., and *Cryptococcus* spp[2,3]. Despite posing a significant threat to public health, fungal infections remain poorly understood and inadequately addressed. Most FDA-approved antifungal agents target the fungal-specific ergosterol in the plasma membrane (polyenes), its biosynthesis (azoles)[4] or disrupt cell wall integrity (echinocandins)[5]. However, the effectiveness of these antifungals has been increasingly compromised by the emergence of drug-resistant species[3,6]. There is an urgent need to enhance existing antifungal compounds and to develop novel therapeutic strategies to combat resistant fungal pathogens.

The fungal plasma membrane and associated proteins are recognized as promising targets for antifungal therapy. Recent advances in structural determination using single particle cryo-electron microscopy (cryo-EM) have provided valuable structural insights into several fungal plasma membrane proteins, the H+-ATPase Pma1[7,8], chitin synthase[9], and β-(1,3)-glucan synthase (GS)[10,11]. However,

[1]Department of Cell Biology and Neuroscience, Rutgers, The State University of New Jersey, Piscataway, NJ, USA. [2]Institute for Quantitative Biomedicine, Rutgers, The State University of New Jersey, Piscataway, NJ, USA. [3]Hackensack Meridian Health-Center for Discovery and Innovation, Nutley, NJ, USA. [4]Computational Biology Department, Carnegie Mellon University, Pittsburgh, PA, USA. [5]Department of Chemistry and Chemical Biology, Rutgers, The State University of New Jersey, Piscataway, NJ, USA. [6]City University of New York-Hunter College, New York, NY, USA. [7]Graduate School of Biochemistry, Rutgers, The State University of New Jersey, Piscataway, NJ, USA. [8]Center for Advanced Biotechnology and Medicine, Rutgers, The State University of New Jersey, Piscataway, NJ, USA. [9]Department of Medical Sciences, Hackensack Meridian School of Medicine, Nutley, NJ, USA. [10]Department of Physics and Astronomy, Rutgers, The State University of New Jersey, Piscataway, NJ, USA. ✉e-mail: david.perlin@hmh-cdi.org; wd157@dls.rutgers.edu

since the structure and function of membrane proteins depend on their membrane environments, determining their structure outside the native membrane context may overlook crucial aspects that are influenced by their lipid surroundings.

GS is responsible for the synthesis of glucans within the fungal cell wall. This enzyme complex consists of a large catalytic subunit encoded by *FKS* genes[12] and a regulatory subunit, GTPase Rho1[13]. The membrane-embedded catalytic subunit is the target of the FDA-approved echinocandin drug class. Despite decades of research, the molecular mechanisms underlying echinocandin inhibition and resistance remain elusive. Our current understanding of echinocandin inhibition of GS largely stems from clinical studies demonstrating that acquired echinocandin resistance is linked to amino acid substitutions in highly conserved hotspot regions of the GS catalytic subunit, encoded by the *FKS1* and *FKS2* genes[5]. Mutagenesis and topology modeling suggest that these hotspot regions may constitute the drug-binding pocket for echinocandins[14]. Intriguingly, the recently reported high-resolution structures of GS[10,11,15] revealed ordered lipid molecules at hotspot regions, suggesting that these lipids are an integral component of GS that may participate in echinocandin interactions with GS. Although current cryo-EM structures have achieved resolutions of 3-4 Å, unambiguous annotation of lipid densities remains challenging and is not consistently achieved across all resolved densities. In addition, some of the integral lipids observed in these structures were likely displaced by detergents during the purification process[15].

Recent studies have also shown that modifications to the lipid composition of the fungal plasma membrane can affect GS susceptibility to echinocandins, providing further evidence that membrane lipids may mediate GS-echinocandin interactions[16–18]. Therefore, investigating the structure and dynamics of GS in its native membrane-embedded state is essential to advancing our understanding of its role in cell wall biosynthesis and its interactions with antifungal drugs.

In this work, we employed a multidisciplinary approach encompassing cryo-electron tomography (cryo-ET), quantitative mass spectrometry, and biochemical, and biophysical analyses to curate a structural atlas of the fungal plasma membrane proteome. We demonstrate that fungal plasma membrane proteins exhibit higher-order organization within the plasma membrane. Echinocandin treatment perturbs the functional organization of membrane proteins, likely by modifying their local lipid environment. Our study elucidates the spatial distribution of prominent fungal plasma membrane proteins in their native environment and offers insights into the emerging role of integral lipid molecules in GS function and their implications for echinocandin action.

## Results

### An integrative workflow toward visual proteomics of the fungal plasma membrane

To enable structural and functional studies of fungal plasma membrane proteins in their native membrane environment, we applied an integrative framework that combines proteomics, microbiological, and biophysical analyses with multimodal bioimaging techniques (Fig. 1). We first performed proteomic analysis on crude membranes isolated from *C. glabrata* spheroplasts to define the protein composition of the fungal plasma membrane fractions. Combining quantitative mass spectrometry with cryo-electron tomography (cryo-ET) and deep-learning based annotation enabled molecular characterization of the fungal plasma membrane landscape in its near-native state. We also performed antifungal susceptibility testing to determine the minimum inhibitory concentration (MIC) of caspofungin (CSF), an echinocandin drug (Supplementary Fig. 1). These results were correlated with micropipette aspiration on intact spheroplasts to assess the effect of CSF on the biophysical properties of the plasma membrane. Additionally, we generated a mutant that is defective in lipid synthesis and employed pharmacological treatment to further validate findings from structural and biophysical studies. The integration of these quantitative and qualitative studies provides a robust, comprehensive framework for characterizing the functional distribution of fungal plasma membrane proteins and exploring the role of the plasma membrane environment in the action of echinocandin drugs.

To prepare plasma membranes for proteomics and structural studies, we subjected *C. glabrata* cells to enzymatic digestion to remove the cell wall. The resulting spheroplasts were then gently ruptured through hypotonic shock to generate crude membrane fractions. This minimal sample preparation protocol preserved the sample in a close-to-native state for subsequent structural and proteomic investigations.

Mass spectrometry analysis of the crude membranes identified 3,905 proteins that represent 74% of the predicted *C. glabrata* proteome, including many proteins that are associated with the nucleus, mitochondrial function, ribosome biogenesis, and other organelles. Our data analysis proceeded in two steps. First, we identified membrane-associated proteins, including both integral plasma membrane proteins and those peripherally associated under specific

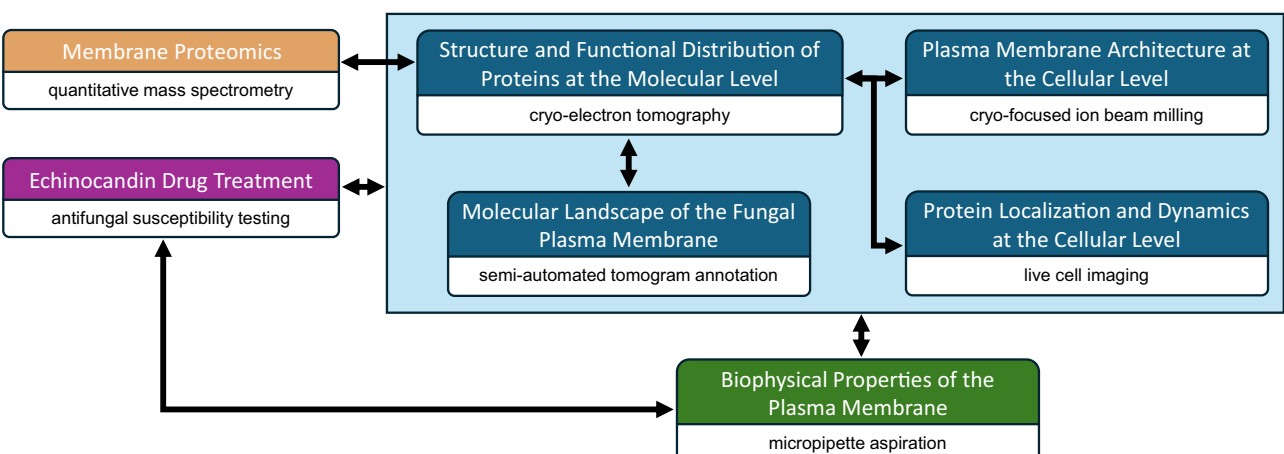

**Fig. 1 | A multidisciplinary approach that enables visual proteomics of the fungal plasma membrane.** Integration of multimodal bioimaging techniques (blue boxes) with quantitative proteomics (orange), microbiological (magenta), and biophysical (green) analyses provides a toolkit to characterize the functional distribution pattern of fungal plasma membrane proteins in their native membrane environment and to dissect the echinocandin mechanism of action. Black connecting arrows indicate the correlation of results generated from different experimental methods.

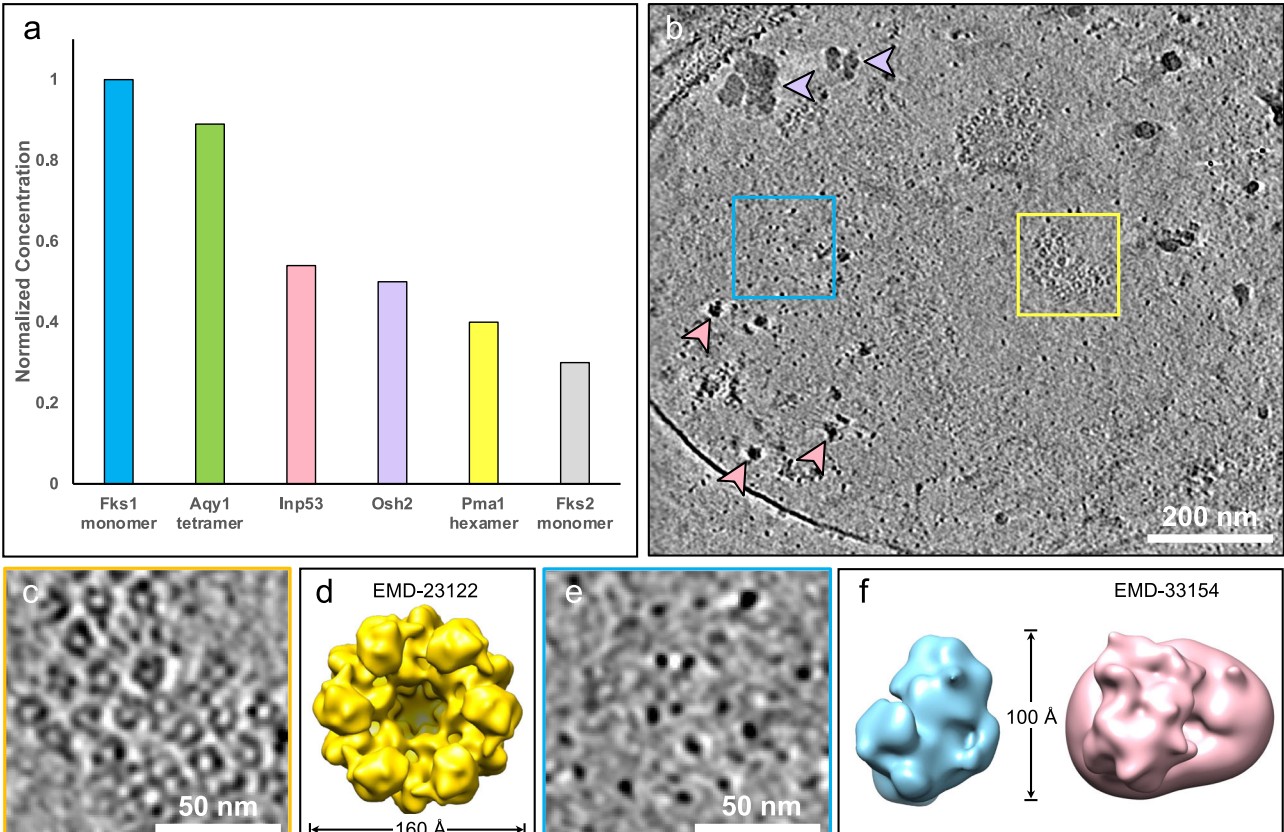

**Fig. 2 | Correlative proteomics analysis and cryo-ET studies characterized the abundance and spatial distribution of abundant fungal plasma membrane proteins in *Candida glabrata*. a** Proteomics analysis of *C. glabrata* crude membrane revealed the relative abundance levels of plasma membrane proteins with molecular weights larger than 100 kDa, normalized to their known oligomeric state. The abundance level of each membrane protein is expressed relative to the level of Fks1. **b** Slice view of a fungal plasma membrane tomogram containing diverse membrane and cytosolic structural features. Purple arrowheads indicate glycogen granules, and pink arrowheads depict ribosomes. Scale bar, 200 nm. This tomogram is representative of a dataset comprising 14 tomograms with high contrast, selected across ~ 100 tomograms acquired from more than three independent samples. **c** Zoomed-in slice view showing rosette-like particles corresponding to Pma1 hexamers boxed in (**b**). Scale bar, 50 nm. **d** Subtomogram average of Pma1 hexamer (EMD-23122)[35]. **e** Zoomed-in slice view showing GS densities boxed in (**b**). Scale bar, 50 nm. **f** Top view of the GS subtomogram average (blue) and the cryo-EM structure of the GS monomer from *Saccharomyces cerevisiae* (pink: EMD-33154)[10] filtered to 20 Å resolution. Source data are provided as a Source Data file.

functional states, and ranked them by relative abundance (Supplementary Table 1). The most abundant proteins identified were involved in pathways related to eisosome assembly, cell wall biosynthesis, and cellular transport. Notably, some top enriched proteins, such as Pst2 and Pst3, quinone oxidoreductases implicated in redox homeostasis[19], are known to localize to multiple subcellular compartments, including the cytoplasm, mitochondria, and the plasma membrane. These proteins are typically recruited to the membrane only under specific functional states and are transiently associated. Another class of highly enriched peripheral membrane proteins included eisosome components such as Lsp1 and Pil1[20]. Their presence further supported that key structural components of the plasma membrane were retained in the crude membrane preparations.

To correlate the plasma membrane proteomic profile with the identity and abundance of protein complexes expected to be observed by cryo-ET, we next focused our analysis on integral plasma membrane proteins. The relative abundance of these proteins was normalized based on their known oligomeric states to better represent the expected number of distinct complexes on the membrane (Fig. 2a). We only included integral membrane proteins or multi-protein complexes with aggregate molecular weights larger than 100 kDa, as proteins below this size threshold may not be detectable by cryo-ET under our imaging conditions. Following this criterion, the top six candidates are Fks1, Fks2, Aqy1, Pma1, Inp53, and Osh2. In *C. glabrata*, the catalytic

subunit of GS is encoded by two homologous genes, *FKS1* and *FKS2*[21]. A single particle cryo-EM structure of GS from *Saccharomyces cerevisiae* suggests that GS exists as a monomer with its catalytic domain in the cytoplasm (PDB ID 7XE4[10]). Aqy1 forms the homotetrameric transmembrane water channel[22]. Inp53 and Osh2 are involved in endocytosis and lipid transport, respectively. Pma1, the fungal H⁺-ATPase, is one of the most abundant plasma membrane proteins and assembles into a functional hexameric complex[7,8]. By combining Fks1 and Fks2 abundance levels, our proteomics analysis revealed a ratio of 3.3 GS monomers to 1 Pma1 hexamer.

Detection of membrane-embedded proteins by cryo-ET depends on the presence of extramembrane domains. Given that Aqy1, Inp53, and Osh2 are largely embedded within the plasma membrane and lack significant extramembrane densities, these proteins may not be readily detectable by cryo-ET[22]. In contrast, Fks1/Fks2 and Pma1 have substantial extramembrane domains and exhibit distinct morphology, making it likely to identify them in cellular tomograms.

Consistent with our proteomics analysis, we observed various membrane structures within our tomograms, including plasma membranes and those derived from cytosolic organelles such as mitochondria and endoplasmic reticulum (ER). We differentiated plasma membranes from other organelle membranes based on overall membrane morphological characteristics and distinctive structural features. For example, rough ER membranes are associated with

ribosomes, while mitochondrial membranes are characterized by the presence of distinct cristae, ATP synthases, and pyruvate dehydrogenase complexes (Supplementary Fig. 2).

Tomograms collected on fungal plasma membranes revealed a diversity of macromolecular complexes (Fig. 2b). The most notable structural features are rosette-like densities with a diameter of approximately 160 Å (Fig. 2b-d; Supplementary Movie S1). These structures correspond to the overall size and morphology of Pma1 hexamers[7,8] and are often present in clusters of varying sizes. Additionally, membrane regions containing Pma1 clusters exhibited noticeable changes in curvature, appearing as membrane depressions in which side views of Pma1 complexes appear as U-shaped protrusions (Supplementary Fig. 3c-e, yellow arrows). Previous structural studies on Pma1 by cryo-EM single particle analysis revealed crystalline-packed lipid molecules within the central cavity of the Pma1 hexamer[7]. These lipids, likely representing ordered sphingolipids, were suggested to stabilize the hexameric conformation. The membrane depression observed at Pma1 cluster-enriched microdomains may reflect the higher local composition of sphingolipids relative to adjacent regions, suggesting a potential role for integral lipids in Pma1 higher-order organization within the membrane.

We also observed oval-shaped densities as another abundant species within our plasma membrane tomograms (Fig. 2b, blue box; Supplementary Movie S1). Given the prevalence of GS in our proteomics analysis, we hypothesized that these densities correspond to GS monomers. The high-resolution cryo-EM structure of the GS monomer exhibits a bulky cytoplasmic domain that measures approximately 100 Å in width and 45 Å in height[10]. Subtomogram averaging of these elongated densities confirms that they exhibit overall shape and dimensions consistent with the structure of the cytosolic region of the GS monomer (Fig. 2e, f; Supplementary Fig. 4). Due to the preferred orientation of plasma membranes on the EM grid, top views of GS were more predominant, resulting in a poorly resolved transmembrane region in the subtomogram average.

Other cellular features were also clearly visible in our tomograms. Large, globular-shaped densities distributed throughout the tomograms correspond to ribosomes (Supplementary Fig. 3, pink arrowheads). We also observed actin filaments (Supplementary Fig. 3, dark blue arrowheads), which have been reported to associate with GS complexes and play a crucial role in maintaining cell wall organization and remodeling[23]. Clover-like structures with diameters of 50–70 nm represent aggregates of glycogen granules that have been previously reported in fungi[24] (Supplementary Fig. 3d, purple arrowheads).

## Fungal plasma membrane proteins segregate into distinct microdomains within the plasma membrane

To elucidate the spatial interrelationships of the membrane protein complexes, we employed a convolutional neural network (CNN)-based annotation approach to identify subcellular features within fungal plasma membrane tomograms[25]. For evaluation of annotation performance, ribosomes and Pma1 hexamers were selected as targets due to their distinct morphology and substantial size. The CNN-based approach demonstrated robust performance in detecting ribosomes, achieving an F1 score of 0.788 (Supplementary Table 2). The relatively high F1 score indicates that the automated annotation of ribosomes achieved a balance between accuracy and sensitivity. Annotation of ribosomes was independent of isosurface, as the performance remained consistent across several isosurface threshold values. In contrast, the annotation of Pma1 hexamers was less effective, with an F1 score of 0.498, and was significantly influenced by the isosurface threshold. Lowering the threshold improved the annotation performance of Pma1 relative to higher isosurface thresholds. Compared to ribosomes, the lower performance of Pma1 annotation and its sensitivity to isosurface settings may be attributed to the inherent challenges of detecting smaller membrane-embedded proteins.

To ensure annotation accuracy, we inspected the results of the automated annotation and manually eliminated obvious false positives and recovered missed structures. We calculated the density of annotated Pma1 and GS complexes in plasma membrane tomograms with high contrast. On average, Pma1 clusters occupied ~15% of the fungal plasma membranes with a density of 264 hexamers per square micrometer of plasma membrane area. GS particles were more densely distributed, with 1415 monomers per square micrometer. The observed GS-to-Pma1 ratio in plasma membrane tomograms is approximately 5.4:1, higher than the 3.3:1 ratio estimated from proteomics data (Fig. 2a), suggesting that GS may preferentially localize to Pma1-enriched microdomains. This potential colocalization is consistent with previous reports that during purification, GS complexes often partition into detergent-resistant, lipid-ordered membrane fractions enriched in sphingolipids and ergosterol[26].

Recent studies have shown that fungal plasma membranes organize into two main membrane microdomains: the MCP (membrane compartment occupied by ATPase Pma1) and the MCC (membrane compartment occupied by arginine permease Can1). MCCs, also called eisosomes, are furrow-like, static membrane structures that segregate the plasma membrane into distinct zones[27,28]. Enriched in Pma1, MCPs are large, network-like structures within the plasma membrane and play a role in dynamic vesicular trafficking[29,30]. We utilized cryo-focused ion beam (cryo-FIB) milling to thin vitrified *C. glabrata* cells and performed cryo-ET imaging on lamellae to examine the membrane microdomain organization within intact cells. Tomograms of cell lamellae displayed invaginations in the plasma membrane that have a width in the range of 40–70 nm and a varying depth of 160–250 nm, characteristic of MCC structures[31] (Supplementary Fig. 5, teal arrowheads; Supplementary Movie S2). We also observed densities corresponding to membrane protein complexes interspersed along the plasma membrane (Supplementary Fig. 5b, orange arrows). However, annotation of these protein complexes was less confident due to the predominance of side views and the crowded cellular environment, which obscured membrane complex structural features.

Given that Pma1 is the landmark protein of MCPs, further characterizing its spatial distribution may provide insights into the overall molecular organization of the plasma membrane. While cryo-FIB milled lamellae preserve native membrane context, the narrow ~200 nm thickness limits the accessible membrane area, making them suboptimal for membrane protein distribution analysis. We therefore performed the analysis using tomograms of crude membrane preparations, which provide larger, more planar membrane surfaces better suited for this purpose.

Tomogram annotation showed that Pma1 hexamers organized into dense, semicrystalline-like clusters within the plasma membrane (Fig. 3a, Supplementary Movie S1). A 3D scatter plot of Pma1 complexes from a representative plasma membrane tomogram revealed variations in the size and shape of distinct Pma1 clusters (Fig. 3b). Nearest neighbor analysis indicated that Pma1 hexamers within the clusters follow a unimodal distribution with a narrow peak at 180 Å, suggesting compact packing of Pma1 complexes (Fig. 3c). To further quantify the distribution of Pma1 clusters within our plasma membrane tomograms, we defined Pma1 clusters based on two criteria: Pma1 hexamers within a cluster are positioned ~160–170 Å apart from one another, and a minimum of four particles are required to constitute a cluster. Using these criteria, we manually annotated Pma1 clusters and validated the outcome by Gaussian Mixture Models (GMM) and *k*-means clustering methods[32,33]. The number of Pma1 clusters determined by both GMM and *k*-means clustering methods closely aligned with our manual assignment of clusters, confirming that Pma1 complexes indeed arrange into distinct clusters within the plasma membrane (Supplementary Table 3).

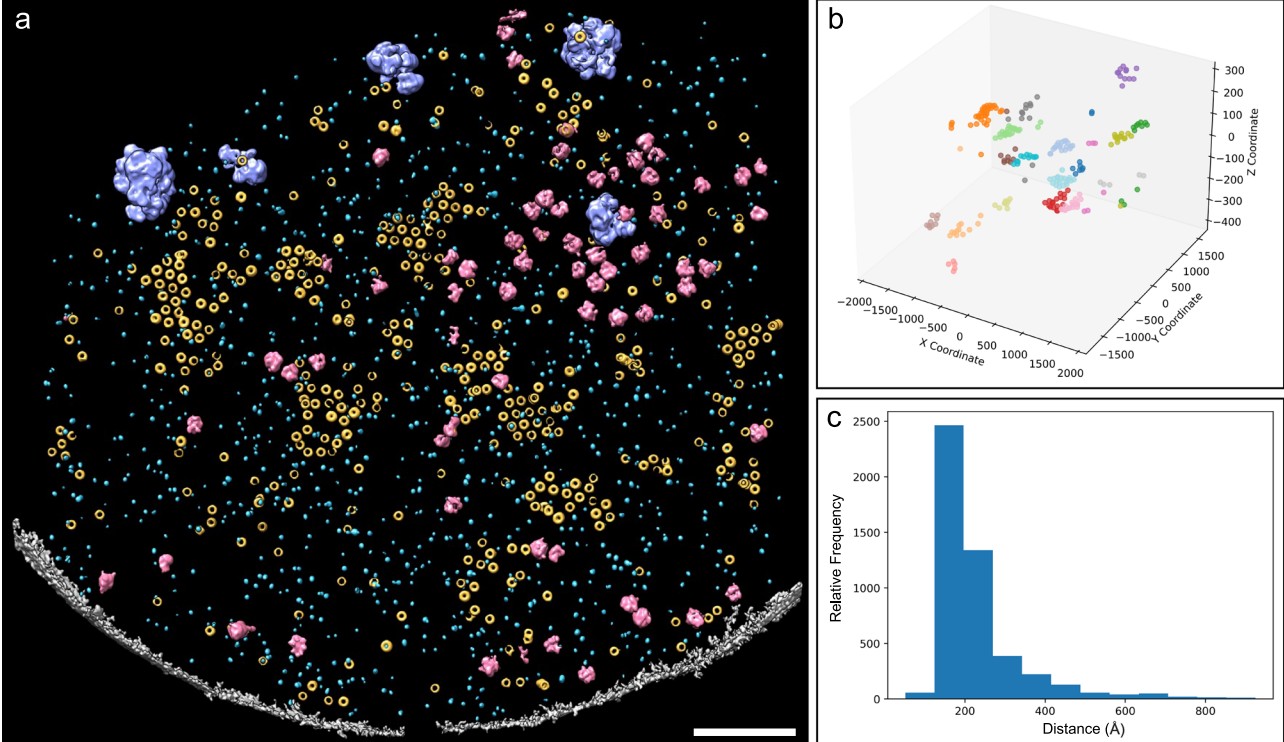

**Fig. 3 | Semi-automated convolutional neural network (CNN)-based annotation revealed the molecular landscape of the fungal plasma membrane.**
**a** Annotation of various structural features in a plasma membrane tomogram. Neural networks were trained independently to recognize Pma1 hexamers (yellow), GS (light blue), ribosomes (pink), and glycogen granules (purple) in the context of the plasma membrane (gray). Scale bar, 200 nm. **b** 3D scatter plot capturing the spatial distribution of individual Pma1 hexamers, represented as dots, within distinct clusters from a representative plasma membrane tomogram. **c** Histogram showing nearest neighbor distance of Pma1 hexamers within clusters in plasma membranes. Source data are provided as a Source Data file.

## Caspofungin treatment perturbs the higher-order organization of plasma membrane protein complexes

GS is the target for the FDA-approved echinocandin and enfumafungin classes of antifungals, which compromise the cell wall integrity and render cells susceptible to osmotic shock. Echinocandin drugs are lipopeptides composed of a cyclic hexapeptide head group and a fatty acyl chain linked to the peptide ring via an amide bond[34]. In high-resolution structures of GS, the putative echinocandin binding pocket, corresponding to the conserved hotspot regions linked to echinocandin resistance, overlaps with the location of integral lipid molecules. We hypothesized that these lipids may directly contribute to GS-echinocandin binding, stabilizing their interactions. We then treated spheroplasts with caspofungin (CSF), the first-in-class echinocandin antifungal, and performed cryo-ET on crude membranes prepared from treated cells.

Our experiment also included a strain that overexpresses Fks1 (herein referred to as KH238) to examine whether an increase in GS abundance levels may affect echinocandin action (Supplementary Fig. 1). In the plasma membrane of the KH238 strain, Pma1 clusters are typically more prevalent and larger, likely due to membrane crowding from overexpression of GS[35].

The most striking observation was changes to the distribution of Pma1 within the plasma membrane of treated cells. Although Pma1 is not the direct pharmacological target of echinocandin antifungals, it is the defining component of MCPs and is readily detectable as hexameric structures in our plasma membrane tomograms. As such, Pma1 serves as a landmark for assessing CSF-induced changes in the spatial organization of fungal plasma membrane proteins.

The CSF-induced Pma1 distribution changes were observed in both the wild type and KH238 cells. Specifically, Pma1 complexes became more dispersed within the plasma membrane (Fig. 4). The average intra-cluster distance and cluster radius of Pma1 hexamers in wild type membranes increased by 1.7-fold after CSF treatment (Fig. 4g, h; Supplementary Tables 3 and 4). Annotation of a representative tomogram of plasma membranes generated from CSF-treated wild type spheroplasts revealed the complete disruption of Pma1 clustering (Supplementary Fig. 6, Supplementary Movie S5). Plasma membranes extracted from KH238 spheroplasts that had been treated with CSF exhibited a 1.9-fold and 2.1-fold increase in the average intra-cluster distance and average cluster radius of Pma1 complexes, respectively. While Pma1 clusters in wild type membranes appeared completely disrupted by CSF treatment, some small Pma1 clusters were retained within KH238 plasma membranes following drug exposure (Fig. 4e, f). In addition, these plasma membranes also exhibited structural deformities, suggesting that CSF exposure impacts the plasma membrane ultrastructure.

While GS appeared more diffusely distributed in CSF-treated crude membrane tomograms, accurate assessment of its spatial distribution in crude membrane tomograms is challenging due to the small size of the complex. To alternatively characterize its distribution at the cellular level, we engineered a strain that expresses Fks1 with an N-terminal yellow fluorescent protein (YFP) tag. This strain exhibited normal echinocandin susceptibility (Supplementary Fig. 1) and GS expression (Supplementary Fig. 7) comparable to the wild type, indicating that the YFP tag does not impair GS function. Aniline blue staining revealed heterogeneous glucan content in the cell wall, as evidenced by a non-uniform fluorescent signal along the cell periphery (Supplementary Fig. 8a–c). In many cells, distinct micron-scale YFP-Fks1 puncta were observed along the cell surface, suggesting that GS complexes localize to specific microdomains within the plasma membrane (Supplementary Fig. 8c). Following CSF treatment, these

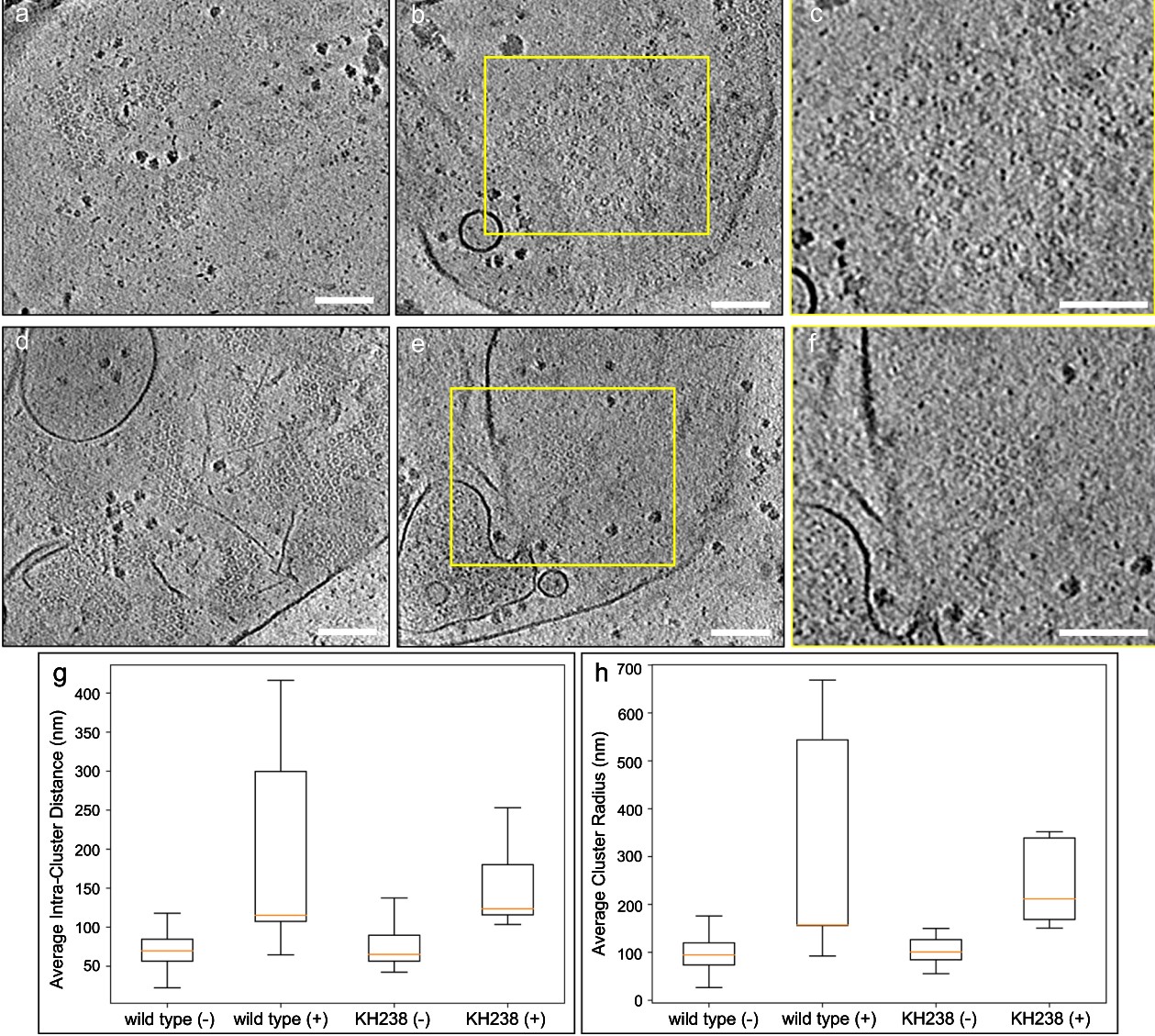

**Fig. 4 | CSF treatment perturbed the spatial distribution of Pma1 within *Candida glabrata* plasma membranes.** Slice views of tomograms of untreated (**a**) and CSF-treated wild type cell membranes (**b**). **c** Zoomed-in view of dispersed Pma1 hexamers boxed in (**b**). Slice views of tomograms of untreated (**d**) and CSF-treated KH238 cell membranes (**e**). **f** Zoomed-in view of a small, tightly packed Pma1 cluster boxed in (**e**). **g** Intra-cluster distance analysis of Pma1 clusters in wild type and KH238 plasma membranes in the absence and presence of CSF treatment. **h** Analysis of average cluster radius in wild type and KH238 plasma membranes in the absence and presence of CSF treatment. wild type (-): $n = 14$; wild type (+): $n = 16$; KH238 (-): $n = 19$; KH238 (+): $n = 7$. (-/+) indicates without or with CSF treatment. Scale bars, 100 nm. Box plots in (g) and (h) show the median (centre line), the 25th and 75th percentiles (bounds of the box), and whiskers extending to 1.5× the interquartile range. Source data are provided as a Source Data file.

puncta were reduced in number and appeared less distinct, indicating disruption of GS complex organization within the plasma membrane (Supplementary Fig. 8c, d). These observations are consistent with the more diffuse GS distribution observed in CSF-treated crude membrane tomograms compared to untreated samples. The general absence of densely packed GS complexes in tomograms may reflect the limited sampling area inherent to cryo-ET, or the low abundance or dynamic nature of GS higher-order organization within the membrane.

Comparison of plasma membrane fractions from untreated and CSF-treated spheroplasts revealed that CSF treatment did not significantly alter the relative abundance of GS and Pma1, or the overall plasma membrane proteome (Supplementary Fig. 9). This suggests that the observed redistribution of Pma1 in CSF-treated membrane is more likely attributable to alterations in the plasma membrane environment induced by CSF treatment, rather than to changes in protein abundance.

## Caspofungin treatment alters the biophysical properties of the plasma membrane

Since CSF perturbed the spatial distribution of Pma1, which is not a direct pharmacological target of the drug, we hypothesized that echinocandin binding to GS may affect the local lipid organization surrounding GS complexes. Such changes could lead to broader alterations in plasma membrane organization and biophysical properties, and be reflected in the changes to the distribution of other plasma membrane proteins, such as Pma1.

To test this hypothesis, we generated spheroplasts from the wild type and a mutant strain with deletion of the *FEN1* gene, which encodes a fatty acid elongase. The Δfen1 strain exhibits defective sphingolipid biosynthesis and a decrease in CSF susceptibility (Supplementary Fig. 1). We then prepared spheroplasts from both strains for micropipette aspiration (MPA) to evaluate the effect of CSF on the biophysical properties of the plasma membrane.

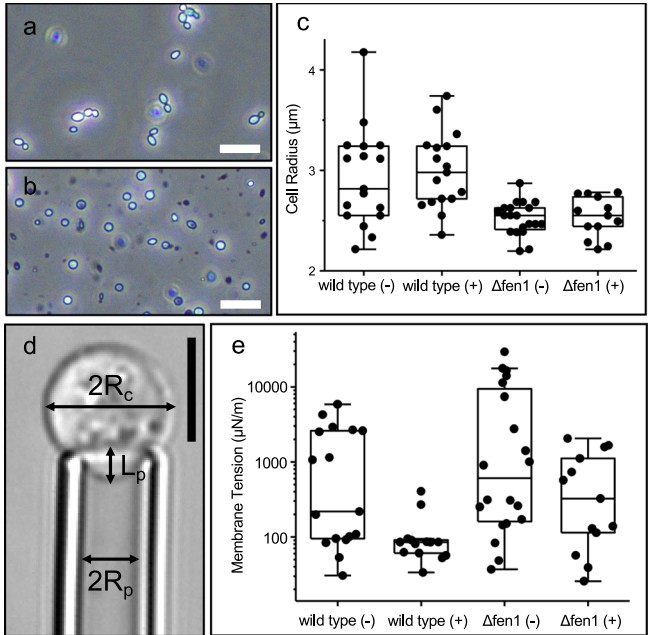

**Fig. 5 | Caspofungin (CSF) treatment altered the biophysical properties of the plasma membrane.** Light microscopy images of *C. glabrata* wild type cells (**a**) and cell wall-less spheroplasts (**b**). Scale bars, 20 μm. **c** Size distribution of wild type and Δfen1 mutant spheroplasts in the absence and the presence of CSF treatment. **d** Micropipette aspiration (MPA) was used to measure the membrane tension of spheroplasts. $R_c$: cell radius; $R_p$: micropipette radius; $L_p$: length of the portion aspirated into the micropipette. Scale bar, 5 μm. **e** The membrane tension of wild type and Δfen1 spheroplasts, with and without CSF treatment. wild type (-): $n = 17$; wild type (+): $n = 14$; Δfen1 (-): $n = 20$; Δfen1 (+): $n = 13$. (-/+) indicates without or with CSF treatment. Box plots in (**c**) and (**e**) show the median (centre line), the 25th and 75th percentiles (bounds of the box), and whiskers extending to 1.5× the interquartile range. Source data are provided as a Source Data file.

Spheroplasts generally exhibit a spherical morphology with an enlarged diameter of 5–6 μm compared to intact cells (Fig. 5a-b). Notably, Δfen1 spheroplasts were slightly smaller than the wild type spheroplasts, likely due to the deficiency in lipid metabolism affecting plasma membrane synthesis or recycling[16,17]. However, treatment with CSF did not affect the spheroplast size significantly in either strain (Fig. 5c).

In the MPA experiment, we measured the membrane tension of spheroplasts before and after CSF treatment by quantifying the degree of membrane deformation in response to stepwise increases in aspiration pressure[36] (Fig. 5d). Compared to wild type spheroplasts, Δfen1 spheroplasts exhibited higher membrane tension under the same micropipette aspiration conditions. This finding indicated that disruption of sphingolipid biosynthesis alters membrane mechanical properties, which can be quantitatively assessed by MPA measurements. Following CSF treatment, the median membrane tension decreased by 2.6-fold in wild type spheroplasts and 1.9-fold in the Δfen1 mutant, respectively (Fig. 5e; Supplementary Movie S3 and S4). These changes likely arise from variations in lipid packing around GS complexes upon CSF binding. Given the high abundance of GS in the plasma membrane, these localized perturbations manifested as a measurable reduction in membrane tension at the cellular level.

We then examined the spatial distribution of Pma1 hexamers in plasma membranes from the Δfen1 strain. Notably, even in the absence of CSF treatment, Δfen1 plasma membranes appear to exhibit rough and textured membrane morphology. Within Δfen1 plasma membranes, Pma1 hexamers were predominantly dispersed rather than clustered (Supplementary Fig. 10). Quantitative mass spectrometry of plasma membrane fractions from wild type and the Δfen1 mutant

revealed comparable relative abundance of Pma1 and Fks1 or the overall plasma membrane proteome (Supplementary Fig. 9). Given that sphingolipids usually reduce membrane fluidity and promote lipid order, the altered distribution of Pma1 in the Δfen1 mutant is likely attributable to increased membrane fluidity resulting from disrupted sphingolipid biosynthesis, which is consistent with the elevated membrane tension observed in Δfen1 spheroplasts in MPA experiments.

Taken together, our findings support the notion that echinocandin binding to GS complexes perturbs the surrounding lipid environment in a manner that propagates across the plasma membrane, leading to spatial reorganization of fungal plasma membrane proteins.

### Disruption of sphingolipid biosynthesis affects CSF susceptibility

Sphingolipid biosynthesis is a complex pathway in which very long-chain fatty acids (VLCFAs) are conjugated to long-chain bases (LCBs), such as dihydrosphingosine (DHS) and phytosphingosine (PHS)[17], to produce sphingolipids with long fatty acyl chains[16,37] (Fig. 6a). Fen1 functions in early steps of VLCFA synthesis. Deletion of *FEN1* reduces VLCFA levels and disrupts the stoichiometric balance between VLCFAs and LCBs.

To further investigate the effect of disruption of the sphingolipid biosynthesis on echinocandin sensitivity, we performed checkerboard assays combining CSF with myriocin, an inhibitor of serine palmitoyltransferase, which catalyzes the first step in LCB synthesis by condensing L-serine and palmitoyl-CoA (Fig. 6b). While no synergy between CSF and myriocin was observed in wild-type cells, myriocin treatment appeared to partially restore CSF susceptibility in the Δfen1 mutant. Since myriocin inhibits LCB synthesis, it may help rebalance the VLCFA-to-LCB ratio toward wild-type levels, thereby altering the membrane lipid environment surrounding GS complexes and enhancing their sensitivity to CSF.

We also performed a checkerboard assay combining CSF with exogenous PHS (Fig. 6c) and observed no synergistic effect between PHS and CSF in either strain. Taken together, the checkerboard experiments suggested that while elevated LCB levels may partially contribute to the reduced CSF susceptibility observed in the Δfen1 mutant, increasing LCB levels alone is not sufficient to alter GS-CSF interactions or susceptibility.

### Inhibition of ergosterol synthesis does not affect echinocandin susceptibility

Ergosterol, a fungal-specific sterol component in the plasma membrane, is essential for maintaining membrane organization and mechanical properties[38], and is the primary target of several FDA-approved antifungals, including fluconazole (FLC). To assess whether ergosterol composition influences echinocandin susceptibility, we performed checkerboard assays combining FLC with CSF or micafungin (MCF) in wild type and Δfen1 mutant strains. Both echinocandins were tested to account for differences in drug chemical structures. There were no synergistic interactions observed between FLC and either CSF or MCF in the wild type strain and a mild synergistic interaction in the Δfen1 mutant strain (Supplementary Fig. 11). This suggested that ergosterol does not significantly influence GS binding or echinocandin susceptibility unless sphingolipid biosynthesis is perturbed.

### Discussion

The fungal plasma membrane and its resident proteins have been intensively studied as potential targets for antifungal therapy. Recent advancements in cryo-EM single particle analysis have laid the groundwork for the structural determination of several key fungal plasma membrane proteins, including Pma1[7,8], chitin synthase[9], and

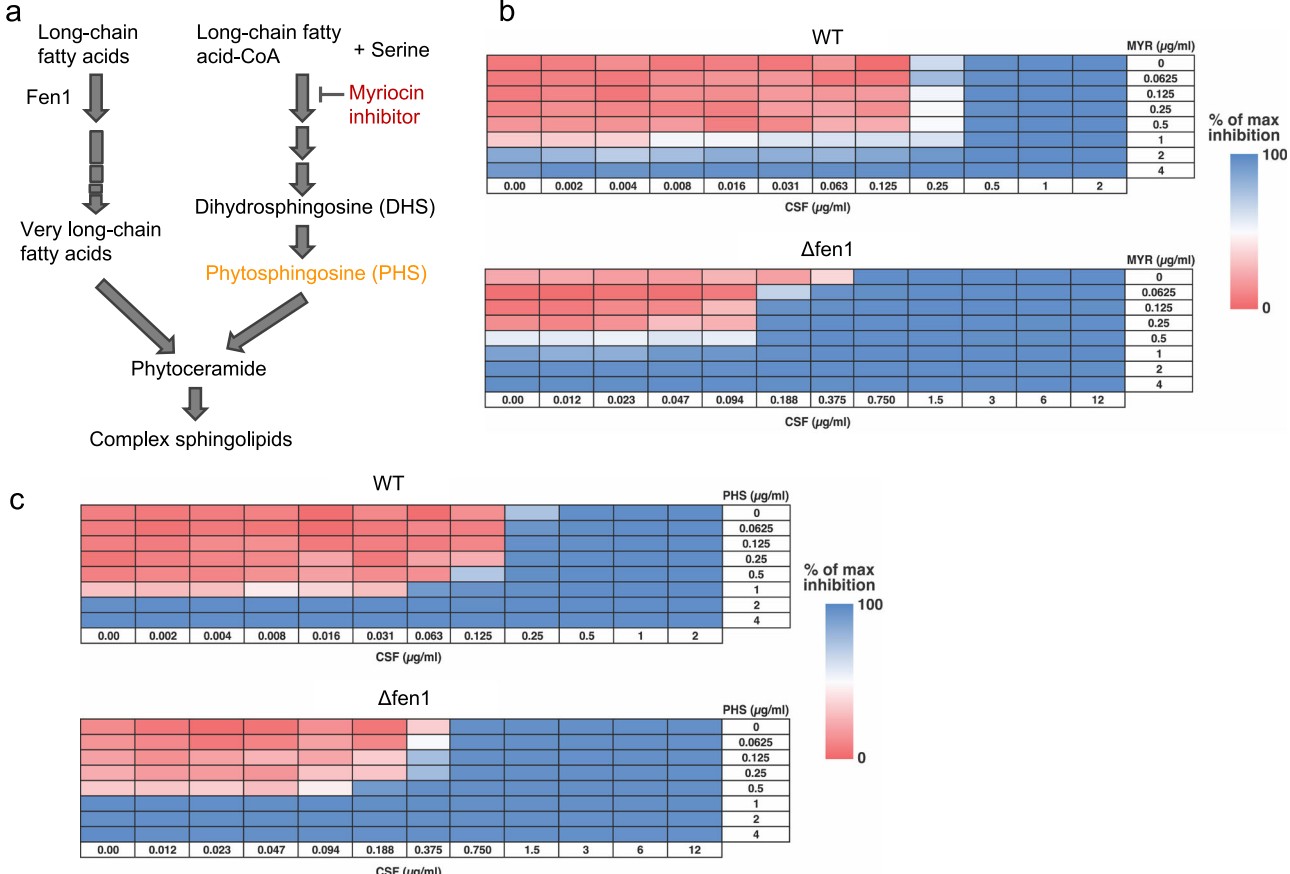

**Fig. 6 | Inhibition of long chain base (LCB) synthesis partially restored caspofungin (CSF) susceptibility in the Δfen1 mutant. a** Schematic of the sphingolipid biosynthesis pathway in fungal cells. **b** Checkerboard assay evaluating the interaction between CSF and myriocin, an inhibitor of LCB synthesis, in wild type and Δfen1 mutant strains. **c** Checkerboard assay assessing the effect of exogenous phytosphingosine (PHS) supplementation on CSF susceptibility. Source data are provided as a Source Data file.

GS[10,11]. These high-resolution structures have provided insights into the molecular mechanisms underlying their central role in fungal physiology and have highlighted the importance of integral membrane lipids in maintaining their structural integrity. Using an integrative multimodal approach centered on cryo-ET, we visualize the molecular landscape of the fungal plasma membrane in its native context.

Our findings demonstrate that prominent fungal plasma membrane proteins exhibit higher-order organization within the plasma membrane. This distribution pattern likely contributes to the cell's ability to perform tightly regulated biological functions in the crowded membrane environment and serves an integral role in fungal infectivity and pathogenicity[39,40].

Our understanding of echinocandin drug action is limited due to the lack of data directly characterizing GS binding to echinocandin antifungals. Topological analysis and mutagenesis studies have mapped the hotspot regions implicated in echinocandin resistance to the extracellular domain of GS[14], supporting their roles as the binding pockets for echinocandins. Binding of echinocandins to these hotspot regions induces conformational changes that extend to the active site on the cytoplasm side, thereby inhibiting glucan biosynthesis. However, experimental efforts to resolve the 3D structures of drug-bound GS have been unsuccessful, suggesting that the interactions between GS and echinocandins could potentially involve other molecular factors in the native cellular environment.

Due to their amphipathic nature, echinocandins have been proposed to insert their hydrophobic tail into the plasma membrane and directly interact with the transmembrane domain of GS complexes. In

our study, CSF exposure altered the biophysical properties of the fungal plasma membrane, and these changes correlated with the perturbed spatial organization of membrane protein complexes. These observations support a lipid-mediated model of echinocandin action. Recent cryo-EM structures of GS revealed integral lipid molecules at the putative echinocandin binding site[10]. These associated lipid molecules regulate the structural and functional dynamics of GS within the plasma membrane. Echinocandin binding to GS may involve these integral lipids at the binding pocket. Echinocandin-GS interaction triggers GS conformational changes that extend to the active site, thereby inhibiting its activity. Additionally, binding of echinocandin to GS disrupts the native lipids, and these disturbances may propagate beyond the drug-target interface, contributing to broader remodeling of membrane microdomains and altered organization of protein complexes on the plasma membrane (Fig. 7). Recent docking simulations suggested that the lipophilic tail of echinocandins can form a ternary complex with these lipid molecules[41], supporting the lipid-mediated model for echinocandin action. We showed that the Δfen1 mutant, which is defective in sphingolipid biosynthesis, exhibited changes in the biophysical properties of the plasma membrane and reduced CSF susceptibility. Further studies will be necessary to determine the molecular details of these interactions and to understand how specific lipid species influence echinocandin inhibition of GS.

In this study, we implemented a robust integrative multimodal workflow and characterized the functional distribution pattern of fungal plasma membrane proteins in their native cellular context. Our work provides novel insights into the role of the plasma membrane in

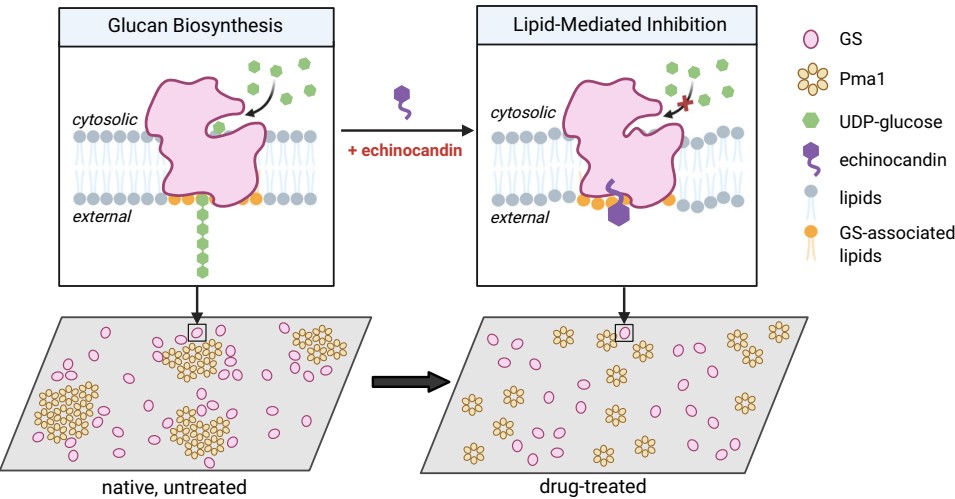

**Fig. 7 | Lipid-mediated model of echinocandin action.** Echinocandin binding to GS, facilitated by its integral lipid molecules at the hotspot regions, disrupts GS function and induces perturbations in membrane organization. Created in BioRender. Dai, W. (2025) https://BioRender.com/yvgu85m.

## Table 1 | *Candida glabrata* strains used in this study

| strain | parent | strain | plasmid | protein expressed | origin |
|---|---|---|---|---|---|
| CBS138 | | wild type strain | | | ATCC |
| KH238 | CBS138 (ATCC2001) | 200989 Δfks1 + pCN-PDC1p-FKS1 | pCN-PDC1 | Fks1 overexpressing | Jiménez-Ortigosa et al.[35] |
| IGCg1 | CBS138 (ATCC2001) | FKS1::NrsR-FKS1p-N-term_YFP-FKS1 | N/A | YFP-Fks1 expressing | this study |
| MVKCg9 | CBS138 (ATCC2001) | ΔFEN1::HygR | N/A | Fen1 deletant mutant | this study |

echinocandin action, opening exciting avenues for the development of more efficacious antifungal strategies.

## Methods

A list of *Candida glabrata* strains used in this study is provided in Table 1.

### Antifungal susceptibility testing and caspofungin treatment

Caspofungin (CSF) acetate (Selleck Chemical) was reconstituted in ultrapure water to yield a stock concentration of 1 mM. The stock was diluted as necessary for antifungal susceptibility tests.

Drug susceptibility microdilution assays were performed according to EUCAST guidelines with modifications[42,43]. Cells were grown and tested in RPMI 1640 media (Life Technologies) containing 2% glucose without sodium bicarbonate, buffered with 165 mM MOPS−HCl to pH 7.0 at room temperature, and filter sterilized. Tested drugs were serially diluted by 100 μl of 2× RPMI in a range of 2× concentrations in flat-bottom 96-well plates (Sarstedt). *C. glabrata* cultures at logarithmic growth phase (OD$_{600}$ = 1–6) were diluted in ultrapure water to OD$_{600}$ = 0.001 and distributed to the drug-containing plates by 100 μl (~1500 CFU/well). Plates were incubated in a moisture-controlling compartment at 37 °C with shaking (150 RPM) in a New Brunswick Innova® 42 incubator (Eppendorf) for 24 h (40 h for slow-growing strains). Plates were scanned at 600 nm using a VERSAmax microplate reader (Molecular Devices). The MICs for CSF were defined as the concentration in the well with OD$_{600}$ < 5% above the optical density for the next (doubled) concentration point. Assays were performed in technical duplicates and two or more biological replicates.

Based on results from the MIC tests, we established the following CSF treatment conditions for each strain: 1 μg/ml for CBS138 and IGCg1, 2 μg/ml for KH238, and 6 μg/ml for MVKCg9. These concentrations are physiologically relevant and enable detectable changes for subsequent multimodal bioimaging studies.

### Preparation of spheroplast and crude membranes

Cultures of *Candida glabrata* cells were grown in YPD (1% yeast extract, 2% peptone, 2% dextrose) or YPD supplemented with 100 μg/mL nourseothricin sulfate (Research Products International) (KH238, plasmid-carrying strain) to mid-logarithmic phase (OD$_{600}$ = 2–3). The cells were harvested by centrifugation (5000 × *g* for 5 min), washed twice with sterile ultrapure water (18.2 MΩ resistance), and incubated with 1% ß-mercaptoethanol for 1 h at 30 °C with gentle shaking (Fisher Scientific). After incubation, the cells were washed twice with water by centrifugation and resuspended in buffer S (1 M sorbitol, 10 mM HEPES-NaOH, pH 6.5) in the ratio of 4 ml of buffer S/gram of wet cells. To remove the cell wall, the cell suspension was digested with lysing enzymes from *Trichoderma harzianum* (Sigma-Aldrich) and Zymolyase 20 T (Zymo Research) at 100 mg and 100 units per gram of wet cells, respectively. The sample was then incubated overnight (14–16 h) at room temperature with gentle shaking. The quality of spheroplasts generated was evaluated under a conventional light microscope.

To generate CSF-treated spheroplasts for cryo-ET and mass spectrometry analyses, intact spheroplasts were treated with CSF at the concentrations described above and incubated for 1 hour at room temperature with gentle shaking. At the end of the incubation, spheroplasts were collected by centrifugation at 3500 *g* at 4 °C and washed twice with buffer S, supplemented with an EDTA-free protease and phosphatase inhibitor cocktail (Roche) with 1 tablet per 10 ml buffer preparations.

To generate crude membrane fractions, the spheroplast pellet was then resuspended in PBS, containing protease inhibitor and phosphatase inhibitor cocktails, for cell disruption. Cell lysis was confirmed by light microscopy. Centrifugation at 20,000 × *g* for 15 min at 4 °C was performed to remove intact cells, spheroplasts, cellular debris, and other microsomal membranes. Crude membranes were resuspended in PBS supplemented with protease inhibitor and phosphatase inhibitor cocktails and used for subsequent mass spectrometry proteomics analysis and cryo-ET structural studies.

## Mass spectrometry and data analysis

We extracted crude membranes from wild type (CBS138), Fks1-overexpressing (KH238), YFP-Fks1 (IGCg1), Δfen1 mutant (MVKCg9), and CSF-treated MVKCg9 cells for mass spectrometry analysis. CBS138 served as the control for the other three strains included in the study. To assess changes in protein abundance following CSF treatment, biological triplicates were analyzed ($n = 3$).

We first ran the samples on an SDS-PAGE. Each gel band was subjected to reduction with 10 mM DTT for 30 min, followed by 60 °C alkylation with 20 mM iodoacetamide for 45 min at room temperature. The sample was kept in the dark and further digested with trypsin (sequencing grade, Thermo Fisher Scientific) and incubated overnight at 37 °C. The digested peptides were extracted twice with 5% formic acid, 60% acetonitrile and dried under vacuum.

For wild type (CBS138), Fks1-overexpressing (KH238), and YFP-Fks1 (IGCg1) cell strains, the resulting peptides were analyzed by LC-MS using Nano LC-MS/MS (Dionex Ultimate 3000 RLSCnano System, Thermo Fisher Scientific) interfaced with Eclipse (Thermo Fisher Scientific). Samples were loaded onto a fused silica trap column Acclaim PepMap 100, 75 μm × 2 cm (Thermo Fisher Scientific) and washed with 0.1% trifluoracetic acid for 5 min with a flow rate of 5 μl/min. The trap column was brought in-line with an analytical column (nanoEase MZ peptide BEH C18, 130 A, 1.7 μm, 75 μm × 250 mm, Waters) for LC-MS/MS. Peptides were fractionated with a flow rate of 300 nl/min using a multistep linear gradient (4–15% Buffer A for 30 min [Buffer A: 0.2% formic acid], then 15–25% Buffer B for 40 min [Buffer B: 0.16% formic acid, 80% acetonitrile], followed by 25–50% Buffer B for 44 min, and 50–90% B for 11 min. For the following run, 4% Buffer B was used for 5 min.

The DIA (Data Independent acquisition) workflow was used for analysis of eluted peptides. The MS scan was set to a resolution of 120,000, with a scan range of 400–1200, and the AGC set to 3E6. An 8 m/z window was used to sequentially isolate (AGC of 4E5 and ion time set to auto) and fragment ions in the C-trap with a relative collision energy of 30. The fragments were recorded with a resolution of 30,000. Raw data were analyzed using the predicted library of the *Candida glabrata* proteome provided on the Uniprot and the FASTA databases for library-free search using DIA NN 1.8.1[44] with recommended settings. The results were filtered for both PEP (an estimate of the posterior error probability for the precursor identification, based on scoring with neural networks) filter <0.01 and PG.Q (Protein Group Q Value) filter <0.01.

To compile a list of only the most abundant fungal plasma membrane proteins present in our sample preparation, we used subcellular localization information available in the Uniprot Knowledgebase, the *Candida* Genome Database (CGD)[45], and the *Saccharomyces* Genome Database (SGD)[46]. These protein candidates are either integral membrane proteins embedded in the cell membrane or associated with the plasma membrane at specific functional states. The multimeric state of the membrane protein candidates was determined by using experimentally derived structures of the protein of interest within the RCSB Protein Data Bank (PDB)[47]. For protein candidates with no oligomerization information, we considered them as monomers. To account for the protein length (i.e., number of amino acids) and the oligomeric state, we normalized the protein group MaxLFQ values of individual proteins to the molecular weight of their functional complex.

To assess potential changes to the relative abundance of Fks1 and Pma1 in the ΔFen1 mutant (MVKCg9) and CSF-treated cells, peptides were analyzed by LC-MS using Nano LC-MS/MS (Neo-Vanquish, ThermoFisher) interfaced with TimsTOF HT (Bruker). Samples were directly loaded on to a PepSep 100, 75 μm × 25 cm (Bruker) and fractionated at 300 nl/min using a segmented linear gradient 2–15% B in 10 min (where A: 0.2% formic acid, and B: 0.2% formic acid, in acetonitrile), 12–32% B in 15 min.

Precursors with m/z between 300 and 1200 were defined in 8 scans (3 quadrupole switches per scan) containing 24 ion mobility steps in an ion mobility range of 0.765 – 1.45(1/k0) with a variable isolation window of 36-41 Th in each step. The acquisition time of each DIA-PASEF scan was set to 85 ms, which led to a total cycle time of around 0.73 second.

The raw data were analyzed using Spectronaut v 20 using recommended settings with *Candida glabrata* reference proteome database for library-free search. Protein group MaxLFQ values reported were used for quantitation and downstream analysis.

## Checkerboard assays

Checkerboard assays were performed in microtiter plates similarly to liquid microdilution assay. To create a two-compound dilution matrix, Agent A in 2X highest concentration in 2X RPMI pH7.0 was serially 2-fold diluted by 7 rows (H to B) in 100 μl, and then 2X of Agent B was similarly serially diluted across columns 12 to 2. Thus, Row A resembled the inhibition curve for Agent B, and Column 1 for Agent A, where well A1 represented a no-drug control. 100 μl of cell suspension in deionized water at OD 0.001 was then added to every well.

After incubation for 24 hours at 37 °C with shaking at 200 rpm, $OD_{600}$ was measured using a VersaMax microplate reader (Molecular devices, CA, USA). Fractional inhibitory concentration (FIC) at ≥90% growth inhibition and FIC indices (FICI) were calculated using the following formulae[48]:

$$\text{FIC Agent A} = MIC_{90}\ \text{Agent A in combination}/MIC_{90}\ \text{Agent A alone} \tag{1}$$

$$\text{FIC Agent B} = MIC_{90}\ \text{Agent B in combination}/MIC_{90}\ \text{Agent B alone} \tag{2}$$

$$\text{FICI} = \text{FIC Agent A} + \text{FIC Agent B} \tag{3}$$

According to guidelines, FICI data were interpreted as 'synergy' (FICI ≤ 0.5), 'no interaction' (FICI > 0.5 but ≤4.0), and 'antagonism' (FICI > 4.0).

## Vitrification of crude membranes and intact yeast cells

Crude membranes obtained from wild type (CBS138), Fks1-overexpressing (KH238), and the sphingolipid-deficient (MVKCg9) strains were mixed with 6 nm BSA gold tracers (Electron Microscopy Sciences) to facilitate tilt series alignment. An aliquot of 3.5 μl of the crude membrane sample was deposited onto freshly glow-discharged Quantifoil R2/2 200-mesh grids (Quantifoil Micro Tools GmbH,) using the Leica EM GP (Leica Microsystems) operated at 20 °C and 95% humidity. Front blotting was performed for 4–6 sec before plunging the EM grid into liquid ethane.

For cryo-FIB experiments, intact *C. glabrata* cells were resuspended in PBS with 10% glycerol and vitrified on Quantifoil Au R2/2 London Finder 300-mesh grids (Quantifoil Micro Tools GmbH) following a similar vitrification procedure as described above for crude membranes. Back-blotting was performed for 10–12 s. All plunge-frozen grids were stored under liquid nitrogen until imaging.

## Cryo-FIB lamellae preparation

A modified protocol was used for micromachining vitrified *C. glabrata* intact cells[49]. Grids were clipped using FIB Autogrids and loaded into an Aquilos 2 dual FIB/SEM instrument (Thermo Fisher Scientific). To improve sample conductivity, sample sputter coating was performed at 10 mA for 25 s. In the microscope chamber, the grids were further deposited with organometallic platinum layers using the in situ GIS to provide a protective coating of the bulk sample. Following GIS

deposition, the grids were sputter-coated for an additional 7 sec at 10 mA to minimize charging artifacts.

Preliminary lamella sites were identified using the MAPS 3 software and AutoTEM 5 was implemented to automate the milling process. A stepwise milling scheme was carried out as follows: rough milling steps of decreasing ion beam current (0.5 nA, 0.3 nA, 0.1 nA) were performed at all lamella sites to remove bulk material above and below the designated lamella position. A final polishing of each lamella was carried out using 50 pA and 10 pA currents to target a nominal thickness of 200–300 nm. Grids containing yeast lamellae were stored under liquid nitrogen until further imaging.

## Cryo-ET and tomogram reconstruction

Micrographs and tilt series of plasma membranes were collected on a Talos Arctica 200 kV microscope (Thermo Fisher Scientific) equipped with a post-column BioQuantum energy filter (slit width of 20 eV) and a K2 direct electron detector. SerialEM was used for automated data collection[50] under the following conditions: 49,000× microscope magnification, spot size 8, 100 μm condenser and objective aperture, and with a nominal defocus range of −4 to −6 μm. The image pixel size was 2.73 Å/pixel. Tilt series were collected continuously from −60° to +60° with 3° step increments and in counting mode with a cumulative dose of 80–100 e⁻/Å².

Tilt series of cryo-lamellae of intact *C. glabrata* cells was acquired on a Titan Krios 300 kV microscope (Thermo Fisher Scientific), equipped with a Selectris X energy filter (set to a slit width 10 eV) and Falcon 4i direct electron detector. SerialEM was used to perform automated data collection at a nominal magnification of 42,000× (pixel size of 3.041 Å/pixel) with a − 6– −7 μm defocus. Tilt series were collected using a dose-symmetric tilt scheme with 3° increments and with a total dose of 90 e⁻/Å².

Raw movie frames were first motion-corrected with MotionCor2 software and tilt series stacks were generated[51]. Tilt series were aligned and reconstructed to generate 3D tomograms using the EMAN tomography workflow[52]. Tomograms were initially reconstructed with a binning factor of 4 to improve contrast.

## Tomogram visualization, subtomogram averaging and annotation

For visualization, 2D slices were generated from 3D tomogram volumes using the IMOD 3dmod interface[53]. We used the EMAN2 convolutional neural network (CNN)-based protocol to annotate the plasma membrane, membrane protein structures, and other subcellular components[25]. Using a high-contrast plasma membrane tomogram, we prepared different training datasets, i.e., positive and negative examples, to train a neural network for each structural feature. For Pma1, 24 positive examples and 132 negative examples were extracted to train the CNN. Due to the smaller size of GS densities, we selected 34 and 189 images for positive and negative samples, respectively. We provided 10 positive and 109 negative examples for ribosomes, and 5 positive and 196 negative samples for glycogen granules. We used images of 64 x 64 pixels for training the CNN on Pma1, GS and ribosomes, and 112 x 112 pixels for glycogen granules. The neural networks were trained for 20 iterations with a learning rate of 0.0001. If the training output appeared suboptimal, we performed additional iterations of training until the accuracy of the CNN learning reached a satisfactory level. The trained neural networks were then applied to annotate structural features within tomograms of plasma membranes generated under similar experimental conditions.

We used manual annotation by an expert human annotator as ground truth to quantitatively assess the performance of the CNN-based annotation. Manual particle picking was conducted for each feature and mapped back the particles to the original coordinates in the tomogram. UCSF Chimera was used to visually compare the manual and CNN-based annotations at various isosurface thresholds[54]. True positives (TP), false negatives (FN) and false positives (FP) were determined and used to calculate the F1 score: TP/[TP + ½(FP + FN)]. Based on these F1 scores, we opted for a lower isosurface setting to maximize the detection of membrane protein targets.

For subtomogram averaging and analysis of GS, a de novo initial model was generated from ~40 GS subtomograms. The initial model was then used for refinement on a larger set of GS subtomograms. We low-pass filtered the cryo-EM structure of *Saccharomyces cerevisiae* Fks1 (EMD-33154)[10] to 20 Å and used it as a reference.

## Statistical analysis of Pma1 and GS density and Pma1 clustering behavior

The density of Pma1 hexamers and GS monomers was calculated as the number of particles normalized to the area of the plasma membrane. In the annotated tomogram, we identified 254 Pma1 particles and 1369 GS particles. To calculate the total membrane area of plasma membranes, we used IMOD to generate a closed contour boundary model of the membrane[53].

To characterize the clustering behavior of Pma1 hexamers in plasma membrane tomograms from different experimental conditions, we manually boxed Pma1 particles in tomograms with sufficient image contrast and generated a list of 3D particle coordinates. We then performed nearest neighbor analysis of Pma1 hexamers in wild type plasma membrane tomograms. Within a Pma1 cluster, the nearest neighbor of a Pma1 hexamer was defined as the particle within the shortest linear distance. We manually estimated the number of distinct clusters present within the plasma membranes, defining a cluster to contain four or more Pma1 hexamers in close proximity to one another with a distance of 160–170 Å. We then used the Gaussian Mixture Model (GMM) and *k*-means clustering algorithms to evaluate our manual assignment of Pma1 clusters[32,33]. While GMM can capture the heterogeneity present in biological samples, *k*-means partitions the data into *k* distinct clusters and optimizes for intra-cluster variance. For GMM, we computed the Akaike Information Criterion (AIC) and Bayesian Information Criterion (BIC) for various clusters and evaluated which models had a lower score. For *k*-means clustering, we determined the optimal number of clusters by assessing the model's inertia, which measures the compactness of the clusters. The number of clusters determined from GMM-based and *k*-means clustering approaches was compared to our manual annotation.

To compare Pma1 distribution patterns across various samples, we performed statistical measures to assess the cluster compactness using intra-cluster distance and cluster radius[55]. Given that the optimal cluster estimation from *k*-means clustering aligned better with our manual annotation results, we calculated the average intra-cluster distance and cluster radius using *k*-means clustering and visualized the results as box plots. We applied the Kruskal-Wallis H test to determine whether these cluster metrics were significantly distinctive across the samples. For both average intra-cluster distance and cluster radius, the Kruskal-Wallis test yielded statistically significant differences with $p$-value $< 0.05$. To facilitate controlled comparisons and mitigate the potential for Type I errors, we proceeded with Dunn's post-hoc test to pinpoint specific pairwise differences between the samples. We then performed power analysis through statistical simulation to ensure the statistical validity of subsequent statistical tests. With 10,000 iterations, we simulated datasets based on the original sample's means and standard deviations with each adjusted for an anticipated effect size of 0.8. This approach showed that the sample sizes were sufficient (estimated power of more than 0.7) to detect differences among the various groups, confirming the reliability of our statistical conclusions.

## Live cell fluorescent imaging

*C. glabrata* cells from CBS138 and IGCg1 strains at the mid-logarithmic growth phase were harvested and resuspended in PBS prior to application on coverslips for live cell fluorescence microscopy. For CSF treatment, CBS138 and IGCg1 cells were first diluted to an $OD_{600}$ of 0.1 prior to treatment with 1 µg/ml CSF. Drug-treated intact cells were incubated at 200 RPM for 1 h at 37 °C, washed with ultrapure water by centrifugation, and resuspended in PBS prior to live cell imaging.

To reduce autofluorescence, 25 mm coverslips (0.17 ± 0.01 mm, Warner Instruments) were pre-cleaned as described previously with slight modifications[56]. Residual solvents were removed by drying using pressured $N_2$. Pre-cleaned coverslips were then plasma cleaned (Harrick Plasma Inc) for 10 min. Plasma-cleaned coverslips were placed into a clean container filled with a water chamber to maintain a humid environment. To immobilize the cells and minimize sample movement, coverslips were coated with 10 mg/ml concanavalin A (Thermo Fisher Scientific) for 5 min. Excess surfactant was gently removed and the coated coverslips were further dried with pressured $N_2$. A volume of 200–500 µl of sample was applied onto the center region of the coverslips and incubated for 5 min to allow the cells to settle prior to visualization.

All microscopy experiments were conducted using a custom-built total internal reflection fluorescence (TIRF) microscope based on the Ti-E inverted microscope, with a high NA CFI-Apo 100X, NA 1.49 objective (Nikon) and an electron multiplying charge-coupled device (EMCCD) camera (iXon Ultra-888; Andor). The microscope was equipped with a 405 nm (OBIS 405 nm LX100 mW; Coherent) and a 488 nm (Genesis MX488-1000 STM; Coherent) lasers for the two-color fluorescence imaging required in this study. The microscope system has ~40% light power delivery efficiency from the laser head to the sample. Differential interference contrast (DIC) imaging was conducted using a white light LED (LDB101F; Prior) and Nikon's DIC modules. Multichannel images were obtained by triggered acquisition schemes, using an acousto-optic tunable filter (AOTF) (AOTFnC-400.650-TN; Quanta-Tech), a transistor-transistor logic (TTL) signal out of the EMCCD camera, a data acquisition card (PCIe-7852R; NI), and Nikon NIS-Elements software.

To stain for β-(1,3)-glucans in the cell wall, yeast cells were incubated in 1.5 mg/ml aniline blue (Ward's Science) for 5 min prior to imaging. Aniline blue was imaged using 405 nm laser excitation and a blue emission band-pass filter (ET475/m; Chroma), while YFP was imaged using 488 nm laser excitation and a yellow emission band-pass filter (ET530/30 m; Chroma). Yeast cells were first inspected in the DIC channel and then switched to the fluorescence channel to optimize imaging parameters (e.g. laser power, illumination angle, camera exposure, etc.) for TIRF imaging. For DIC imaging, the exposure time was typically set to 50 ms. For imaging aniline blue, the 405 nm laser was used with a power setting of 10 mW, coupled with an exposure time of 500 ms. For imaging YFP, the 488 nm laser was used with the power setting set to 10 mW, coupled with an exposure time of 500 ms. The electron multiplication (EM) gain of the EMCCD camera was set to 50 and 230 for DIC and fluorescence imaging, respectively. The Perfect Focus System (Nikon) was implemented to actively stabilize focus drift during image acquisition. Digital image analysis was performed using Fiji (ImageJ) software[57].

## Measurement of membrane tension by micropipette aspiration (MPA)

Micropipettes were pulled from glass capillaries using a pipette puller (PUL-1000, World Precision Instruments (WPI)). The pipette tip was cut to an opening diameter smaller than the average spheroplast radius, ranging from 1.5 to 2 µm.

Micropipette aspiration (MPA) and imaging were performed on a Ti2-A inverted fluorescent microscope (Nikon) equipped with a motorized stage and two motorized micromanipulators (PatchPro-5000, Scientifica). *C. glabrata* spheroplasts were maintained in Buffer S, an osmotically stabilizing buffer, to prevent lysis. A micropipette was filled with the same buffer used for spheroplasts using a MICROFIL needle (WPI) and then mounted onto a micromanipulator connected to the pressure control (LU-FEZ-N069, Fluigent).

The zero pressure of the system was calibrated before each MPA experiment, using a dilute solution of small particles. The zero pressure (P0) was set according to the point when the particles undergo random Brownian motion in the micropipette. After calibration of the aspiration pressure, 100–200 µl sample of the spheroplast sample was loaded onto the center of a glass-bottom dish (ES56291, Azer Scientific) and ultrapure water was added to the edge of the dish to minimize sample evaporation. A calibrated micropipette was then moved to a spheroplast with a diameter of approximately 5 µm. During aspiration measurements, sequential stepwise suction pressures were applied to deform the spheroplast. Aspiration pressure was gradually increased every 20–30 s. The membrane tension σ was calculated as $\sigma = \Delta P \cdot R_p / [2(1 - R_p/R_c)]$ where $R_p$ and $R_c$ are the micropipette and cell radius, respectively, and ΔP is the aspiration pressure at which the length of the spheroplast aspirated into the micropipette is closest to $R_p$. Deformation of the spheroplast membrane was recorded using a 60X objective at Hz (ORCA-Flash 4.0, Hamamatsu) through transmitted light imaging. For drug treatment experiments, spheroplasts were treated with 1 µg/ml CSF for 15 min prior to micropipette aspiration and imaging. Fiji (ImageJ) was used to track the shape parameters ($L_p$, $R_p$, $R_c$) of the aspirated spheroplasts.

## Reporting summary

Further information on research design is available in the Nature Portfolio Reporting Summary linked to this article.

## Data availability

Tomograms of plasma membranes from untreated wild type cells and CSF-treated cells have been deposited in the EMDataBank under accession codes EMD-45105 and EMD-45106, respectively. Mass spectrometry data for the wild-type strain (CBS138), Fks1-overexpression strain (KH238), YFP-Fks1 strain (IGCg1), and the Δfen1 mutant strain (MVKCg9), both untreated and CSF-treated, have been deposited in MassIVE under accession number MSV000098079 [https://massive.ucsd.edu/ProteoSAFe/dataset.jsp?task=a02613ee6a46406d933560d130e61828]. Source data are provided with this paper.

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

## Acknowledgements

We thank Jason T. Kaelber at the Rutgers CryoEM & Nanoimaging Facility (RCNF) for their technical assistance in data collection. A portion of this research was supported by NIH grant U24 GM139168 and performed at the Midwest Center for Cryo-ET (MCCET) and the Cryo-EM Research Center in the Department of Biochemistry at the University of Wisconsin. We would like to thank Muyuan Chen for his technical support with EMAN2. The authors acknowledge NIH S10 OD036226, which supported the purchase of the TimsTOF HT used for mass spectrometry data acquisition W.D. was supported by a Busch Biomedical Research Grant from Rutgers University. J. J., Y-K. L. and W. D. were partially supported by NSF CAREER MCB-2046180. J.J. was also partially supported by a Rutgers University Presidential Graduate Fellowship awarded by the Rutgers Institute for Quantitative Biomedicine. Y.H., E.S. and D.S.P. were partially supported by NIH R01AI109025. X.Z. and M.X. were partially supported by NIH R01GM134020 and NSF CAREER DBI-2238093. S-H.L. was supported by DOE DE-SC0019313. H.W. and Z.S. were supported by R35GM147027 and R21DA056322.

## Author contributions

J.J., M.V.K., D.S.P. and W.D. designed experiments. J.J., M.V.K., A.P., X.Z., J.C., H.W., G.L., Y-K.L., C.P. and N.J. performed experiments and data analysis under the supervision and guidance of Z.S., S-H.L., M.X., D.S.P., and W.D. C.Z., H.Z. performed the quantitative mass spectrometry analysis. M.V.K, Y.H. and E.S. designed and generated the strains used in this study. M.V.K. and E.S. performed the checkerboard assays. J.J. and W.D. wrote the manuscript with input from all authors.

## Competing interests

The authors declare no competing interests.
