## [Transparent Peer Review file · Nature Communications]

Molecular Landscape of the Fungal Plasma Membrane and Implications for Antifungal Action

Corresponding Author: Dr Wei Dai

Version 0:

Reviewer comments:

Reviewer #1

(Remarks to the Author)

In this study, Jiang et al. employed an integrative approach, combining cryo-EM, live cell imaging, mass spectrometry, and micropipette aspiration, to examine the distribution of glucan synthase (GS) and proton pump Pma1 within the fungal plasma membrane. They found that these two membrane proteins exhibit heterogeneous distribution within the membrane, and that the antifungal drug caspofungin (CSF) disrupts their higher-order organization. This work suggests the importance of plasma membrane context for echinocandin inhibition of GS. While the results presented in this manuscript offer valuable insights into the local organization of membrane proteins, certain conclusions warrant further consideration.

Specific comments:

- 1) The authors observed that Pma1 and GS are heterogeneously distributed in the plasma membrane, and treatment with CSF disrupts this higher-order distribution. While it's conceivable that changes in membrane protein distribution are linked to CSF inhibition, the cause-and-effect relationship remains unclear. The authors propose a model where CSF insertion modifies the local membrane environment, thereby altering Pma1 and GS organization. However, this model lacks direct experimental evidence. Further, the manuscript does not elucidate how changes in membrane properties would facilitate CSF's interaction with GS. In experiments with the delta-fen1 strain (unable to synthesize sphingolipids), Pma1 is predominantly dispersed, similar to CSF-treated wildtype cells. However, this strain is less susceptible to CSF. Conversely, when Fks1 is overexpressed (leading to increased GS abundance and polysaccharide synthesis), Pma1 clusters are more prevalent and larger. These results suggest that changes in membrane protein distribution might be an effect caused by altered lipid composition, which in turn, could be caused by CSF inhibition of GS.
- 2) The investigation of Pma1 and GS distribution utilized crude membranes prepared from spheroplasts. It's worth noting that the enzymatic treatment and membrane isolation process could have significantly altered the physical and biochemical properties of the membrane. While the authors conducted cryo-ET study of FIB-milled lamellae of yeast cells, a potentially superior approach for studying protein distributions in native membranes, they did not examine Pma1 or GS distributions within these lamellae, nor did they compare them to those observed in isolated membranes.
- 3) In the cryo-ET reconstructions, Pma1 and GS are annotated only at flat surfaces where membrane density is not resolved (due to missing information along the Z-axis). Notably, no Pma1 or GS can be identified at membrane edges where membrane density is well-resolved. This observation could be due to membrane squashing within the thin ice layer caused by surface tension, exposing membrane surfaces to the air/water interface. These factors could significantly alter membrane properties, highlighting the need to interpret membrane protein distributions obtained from cryo-FIB milled lamellae.
- 4) In this study, the annotation of membrane proteins in cryo-ET reconstructions relies on structural features like shape and size. While effective for large protein complexes with distinct features (e.g., ribosomes), this method might not be suitable for smaller proteins like GS. To validate this annotation approach, a positive control such as antibody labeling of GS or Pma1 for cryo-ET analysis would be necessary.
- 5) The apparent even dispersion of GS in the cryo-ET reconstructions, regardless of CST treatment, contradicts the manuscript's conclusions. Moreover, the accuracy of GS annotation is questionable due to its small size.
- 6) The proteomics analysis indicates that Pma1 and GS are significantly less abundant than many other membrane proteins; for example, Pst2 is over 200 times more abundant than Pma1 and GS. However, in the cryo-ET reconstructions, Pma1 and GS appear abundant. This discrepancy is concerning, as it seems improbable to accommodate several hundred times more proteins within the same membranes.
- 7) It will be useful to also include proteomics analysis of CST-treated membranes.
- 8) Regarding the live cell fluorescent imaging experiments, additional details are required, particularly concerning CST treatment protocols. It is also essential to quantify YFP-Fks1 expression levels across different experimental conditions.

Reviewer #2

(Remarks to the Author)

Reviewer #3

(Remarks to the Author)

The present study by Jang et al studies the “landscape” of proteins in the yeast plasma membrane. How are they organized and what affects the organization?

By proteomics analysis the abundance of proteins in *C. glabrata* was estimated, with focus on proteins above 100 kDa. Among the candidates two proteins were selected, GS and PMA1, Cryo-ET revealed that both GS and PMA1 are heterogeneously distributed in the PM, but in two different types of microdomains.

PMA1 distribution may provide insights into the overall molecular organization of the plasma-membrane. Tomogram annotation showed that Pma1 hexamers organized into dense semicrystalline-like clusters within the plasma membrane. These clusters of PMA1 were computationally annotated to analyse the spatial distribution.

Biophysical data demonstrate that the echinocandin class drug caspofungin (CSF) treatment induces changes in plasma membrane properties, specifically surface tension.

Also, CSF treatment resulted in less distinct GS and PMA1 protein clusters. Based on these data it is concluded that echinocandins interact with and modify the fungal plasma membrane, and these changes in membrane properties lead to spatial rearrangement of fungal plasma membrane proteins.

Comment:

MP proteins are considered potential targets for new antifungal treatments. This is also the rationale for the current study.

PMA1 proteins are found to form the rosette like structures (hexamers) as seen before in cryo structures of PMA1. (Heit et al. Structure of the hexameric fungal plasma membrane proton pump in its autoinhibited state. (2021) *Sci Adv.* doi: 10.1126/sciadv.abj5255.); Zhao et al (2021). Structure and activation mechanism of the hexameric plasma membrane H⁺-ATPase. *Nat Commun.* doi: 10.1038/s41467-021-26782-y.).

In the present work, PMA1 proteins are found in subdomains appearing to induce curvature, it is however not demonstrated if this of biological importance.

The workflow described in the paper is of great value to the scientific field and might have applications in developing new drugs in combination with other studies.

How this paper might help lead to the direct biological understanding the mode of action of echinocandins might be further away, and the biological question answered seems a bit far-fetched. The work documents very well the organization of the plasma membrane in response to caspofungin treatment, but the implication for antifungal action remains poorly understood.

We don't find it surprising that the membrane surface tension is affected when treating the spheroblasts with CSF, as CSF has detergent-like properties (hydrophilic hexa-peptide headgroup and hydrophobic aliphatic tailgroup) and as spheroblasts generally are known to be very fragile. What is the effect of adding a simpler amphiphilic molecule? We suggest including a control with a relevant detergent that will integrate in the membrane but not specifically target GS.

The authors use a mutant yeast strain to show that certain lipids are important for CSF activity (mainly sphingolipids). However, we lack some functional data in this study to understand the effect of CSF on the two proteins. We suggest including functional data on PMA1 and GS to test the effect of CSF treatment.

Quality of work

The experiments seem to be performed thoroughly with an appropriate number of controls and repetitions.

The manuscript is well documented with a comprehensive supplementary data.

The manuscript is well written in a concise manner.

Reviewer #4

(Remarks to the Author)

The paper by Jiang et al. describes the perturbation of fungal plasma membrane upon treatment with the antifungal drug caspofungin. The authors found that caspofungin perturbs the distribution of the polysaccharide beta-glucans and the

plasma membrane proton pump Pma1. Overall, the methodology used is the first of its kind. The results are well organized, well-presented and well-discussed. Few points are below.

1) Fen1 is an enzyme for the synthesis of very long chain fatty acids, which are fatty acids used to make various lipids, including sphingolipids. Thus, a mutant lacking Fen1 will have a defect in the synthesis of lipids, including sphingolipids, containing very long chain fatty acids. Thus, it is possible that the defects observed in the paper by the authors in the delta-fen1 may be due to the lack of the lipids other than sphingolipids. Please clarify these issues.

2) Have the authors considered using different mutants more specifically involved in the sphingolipid (or ergosterol) pathways? Alternatively, drugs that inhibits specifically enzymes involved in the synthesis of sphingolipids exist. Have the authors considered using them in combination with caspofungin?

3) Plasma membrane perturbations are clearly visible upon caspofungin treatment. But ergosterol is a key component of the fungal plasma membrane. Have the authors considered using in their assay an antifungal diminishing ergosterol synthesis (e.g. fluconazole) or altering ergosterol function(e.g. amphotericin b)? These drugs may or may not cause perturbations in the plasma membrane similar to those caused by caspofungin (e.g. beta-glucan/Pma1).

Reviewer #5

(Remarks to the Author)

Version 1:

Reviewer comments:

Reviewer #1

(Remarks to the Author)

This revised manuscript addresses the reviewer's all comments.

Reviewer #2

(Remarks to the Author)

Reviewer #4

(Remarks to the Author)

With this revision, the authors responded to my comments in a satisfactory manner and I do not have additional comments or questions.

Molecular Landscape of the Fungal Plasma Membrane and Implications for Antifungal Action

Jennifer Jiang, Mikhail V. Keniya, Anusha Puri, Xueying Zhan, Jeff Cheng, Huan Wang, Gigi Lin, Yun-Kyung Lee, Nora Jaber, Caifeng Zhao, Cynthia Pang, Yasmine Hassoun, Haiyan Zheng, Erika Shor, Zheng Shi, Sang-Hyuk Lee, Min Xu, David S. Perlin, Wei Dai

Responses to reviewers

We thank all reviewers for their thoughtful comments and constructive suggestions. In response, we have made substantial revisions to the manuscript, focusing on two major areas: First, we conducted additional microbiological analyses combining echinocandin treatment with pharmacological inhibitors targeting sphingolipid and ergosterol biosynthesis. These experiments were designed to strengthen the mechanistic link between membrane composition/environment and antifungal drug susceptibility. Second, we performed quantitative mass spectrometry on membrane preparations used in our structural studies. This analysis allowed us to assess whether the structural changes observed by cryo-electron tomography (cryoET) reflect alterations in the membrane environment rather than changes in membrane protein abundance. In light of these new data, we have added three co-authors who contributed significantly to the additional experiments and data interpretation.

Below, we provide a point-by-point response to each reviewer's comment.

Reviewer #1:

In this study, Jiang et al. employed an integrative approach, combining cryo-EM, live cell imaging, mass spectrometry, and micropipette aspiration, to examine the distribution of glucan synthase (GS) and proton pump Pma1 within the fungal plasma membrane. They found that these two membrane proteins exhibit heterogeneous distribution within the membrane, and that the antifungal drug caspofungin (CSF) disrupts their higher-order organization. This work suggests the importance of plasma membrane context for echinocandin inhibition of GS. While the results presented in this manuscript offer valuable insights into the local organization of membrane proteins, certain conclusions warrant further consideration.

Specific comments:

1) The authors observed that Pma1 and GS are heterogeneously distributed in the plasma membrane, and treatment with CSF disrupts this higher-order distribution. While it's conceivable that changes in membrane protein distribution are linked to CSF inhibition, the cause-and-effect relationship remains unclear. The authors propose a model where CSF insertion modifies the local membrane environment, thereby altering Pma1 and GS organization. However, this model lacks direct experimental evidence. Further, the manuscript does not elucidate how changes in membrane properties would facilitate CSF's interaction with GS. In experiments with the delta-fen1 strain (unable to synthesize sphingolipids), Pma1 is predominantly dispersed, similar to CSF-treated wildtype cells. However, this strain is less susceptible to CSF. Conversely, when Fks1 is overexpressed (leading to increased GS abundance and polysaccharide synthesis), Pma1 clusters are more prevalent and larger. These results suggest that changes in

membrane protein distribution might be an effect caused by altered lipid composition, which in turn, could be caused by CSF inhibition of GS.

We thank the reviewer for this thoughtful and constructive comment. We agree that a direct relationship between CSF treatment, GS inhibition, and membrane protein remodeling warrants further clarification. Recent high-resolution cryo-EM structures of GS revealed that the potential echinocandin-binding pocket co-localizes with integral lipids within the transmembrane region (Hu et al., DOI: 10.1038/s41586-023-05856-5). Notably, these lipid densities varied depending on the detergent used for GS purification and correlated with differences in drug susceptibility. These findings support a role for integral membrane lipids in GS-echinocandin binding, although these studies are inherently limited by the absence of a native membrane context.

As the reviewer points out, both lipid composition and protein abundance could affect the membrane protein distribution observed in our cryoET analyses. However, several lines of evidence from our data support the model that CSF treatment induces reorganization of the membrane lipid environment, which in turn drives redistribution of membrane protein complexes.

- First, micropipette aspiration experiments revealed that CSF-treatment altered membrane tension. These measurements were done within 15 minutes of CSF exposure, consistent with an immediate biophysical effect on the plasma membrane. These rapid changes are unlikely to result from transcriptional or translational regulation, suggesting that CSF treatment, at least in part, directly affects membrane properties.
- In the revised manuscript, we added quantitative mass spectrometry on crude membrane fractions from CSF-treated, Δ fen1, and FKS1-overexpression strains. These data allow us to evaluate whether changes in membrane protein distribution upon CSF treatment could be caused by alterations in protein abundance (new **Supplementary Figs 7 and 9**). These analyses showed that Δ fen1 mutants have comparable GS abundance to wild-type cells, indicating that their reduced CSF susceptibility is not due to reduced GS expression. Importantly, CSF treatment did not significantly alter the relative abundance of GS or Pma1 in wild type cells (new **Supplementary Fig. 9**). These results suggest that the large-scale reorganization of GS and Pma1 observed in our studies is unlikely to stem from changes in protein abundance and more likely reflects changes in the biophysical properties of the membrane.
- We also conducted drug interaction studies using inhibitors of sphingolipid and ergosterol biosynthesis to determine whether changes in membrane lipid composition affect GS-echinocandin interactions or echinocandin susceptibility.
 - We combined CSF treatment with either myriocin (an inhibitor of serine palmitoyltransferase) or phytosphingosine (PHS) supplementation to perturb sphingolipid biosynthesis (new **Fig. 6**, Lines 440–460). In both cases, no significant pharmacological interactions with CSF were detected in wild-type cells, although it seems that inhibition of long-chain base synthesis by myriocin partially recovers echinocandin susceptibility in the Δ fen1 mutant. Since myriocin inhibits LCB synthesis, it may help rebalance the VLCFA-to-LCB ratio toward wild-type levels, thereby altering the membrane lipid

environment surrounding GS complexes and enhancing their sensitivity to CSF.

- We also evaluated the effect of fluconazole (FLC), an inhibitor of ergosterol synthesis, in combination with CSF or micafungin (MCF) in wild-type and Δ fen1 strains (new **Supplementary Fig. 11**, Lines 470-481). Results showed that there were no synergistic interactions observed between FLC and either CSF or MCF in the wild type strain and a mild synergistic interaction in the Δ fen1 mutant strain. This suggests that ergosterol does not significantly influence GS binding or echinocandin susceptibility unless sphingolipid biosynthesis is perturbed.

Together, these new experiments indicate that global changes in membrane lipid composition do not significantly affect GS-echinocandin interaction, unless the native lipid environment has been disturbed, as in the Δ fen1 mutant. These findings support a model in which CSF-GS interactions lead to remodeling of the native lipid environment, which in turn alters the distribution and organization of membrane proteins.

2) The investigation of Pma1 and GS distribution utilized crude membranes prepared from spheroplasts. It's worth noting that the enzymatic treatment and membrane isolation process could have significantly altered the physical and biochemical properties of the membrane. While the authors conducted cryo-ET study of FIB-milled lamellae of yeast cells, a potentially superior approach for studying protein distributions in native membranes, they did not examine Pma1 or GS distributions within these lamellae, nor did they compare them to those observed in isolated membranes.

We agree with the reviewer that FIB-milled lamellae offer the advantage of preserving native membrane architecture and minimizing potential artifacts introduced during sample preparation. However, for this study, we used crude membrane preparations because they currently provide several important advantages for mapping of membrane protein distributions.

First, data yield and analyzable membrane area are significantly larger in crude membrane tomograms. A tomogram of a crude membrane preparation typically covers approximately $1 \mu\text{m}^2$ of membrane surface (and up to $\sim 2 \mu\text{m}^2$ when forming a vesicle), whereas a tomogram from a FIB-thinned lamella contains $\sim 0.3\text{--}0.6 \mu\text{m}^2$ of plasma membrane, even under optimal milling geometry. This reduced sampling area and the low-throughput nature of lamella preparation pose practical limitations for large-scale protein spatial analysis.

Second, the narrow milling thickness of lamellae ($\sim 200 \text{ nm}$) restricts the field of view and can truncate larger membrane protein clusters, making it suboptimal for evaluating higher-order distribution. In contrast, crude membranes provide open, planar views that are well suited for spatial organization analyses.

Third, annotation is more challenging in lamella tomograms. Lamella tomograms often capture side views of membrane protein complexes, where structural signatures are less distinct and vary depending on the position of the tomographic slice relative to the complex. Furthermore, the crowded intracellular environment introduces substantial background complexity that further reduces annotation accuracy and confidence. To

illustrate this point, we included examples of annotated putative membrane proteins in a lamella tomogram in **Supplementary Fig. 5B**.

For these reasons, we focused our quantitative spatial analyses on crude membrane preparations, while using FIB-milled lamellae primarily to assess large-scale membrane architecture. We have clarified this rationale in the revised manuscript (Lines 270-272, 275-279).

3) In the cryo-ET reconstructions, Pma1 and GS are annotated only at flat surfaces where membrane density is not resolved (due to missing information along the Z-axis). Notably, no Pma1 or GS can be identified at membrane edges where membrane density is well-resolved. This observation could be due to membrane squashing within the thin ice layer caused by surface tension, exposing membrane surfaces to the air/water interface. These factors could significantly alter membrane properties, highlighting the need to interpret membrane protein distributions obtained from cryo-FIB milled lamellae.

Crude membrane preparations tend to settle flat on EM grids, which indeed introduces a preferential sampling of top views and limits structural information along the Z-axis. As the reviewer noted, protein complexes can be easily annotated on the flatter membrane surfaces where extra-membrane density is more readily interpretable, while edges tend to be underrepresented. Nonetheless, in our revised manuscript, we now include representative tomographic slices showing cross-sectional (side) views of both Pma1 and putative GS complexes from crude membrane tomograms (**Supplementary Fig. 3**), demonstrating that these orientations, while less frequent, are present.

To address the reviewer's concern about potential membrane deformation caused by air-water interface - while some deformation during blotting and vitrification is inevitable, integral membrane proteins are less susceptible to exposure at the interface compared to soluble proteins, due to their embedding in the bilayer. We also note that ice thickness plays a critical role. Membrane deformation becomes more prominent only when the ice is excessively thin. Most of our data were acquired under conditions that maintained sufficient ice thickness to preserve membrane and protein integrity. In our tomograms, orthogonal views of crude membrane tomograms show that membrane complexes are often positioned away from the ice surface, which is frequently identifiable by the presence of some ice contaminants.

4) In this study, the annotation of membrane proteins in cryo-ET reconstructions relies on structural features like shape and size. While effective for large protein complexes with distinct features (e.g., ribosomes), this method might not be suitable for smaller proteins like GS. To validate this annotation approach, a positive control such as antibody labeling of GS or Pma1 for cryo-ET analysis would be necessary.

We thank the reviewer for this helpful suggestion. We did attempt antibody labeling of both GS and Pma1; however, the results were inconclusive due to limitations in antibody specificity, low labeling efficiency, and partial protein degradation during

experiments. Although immunolabeling is a powerful tool in some contexts, it remains technically challenging in cryo-ET studies of small membrane proteins.

To support the accuracy of annotation, we used high-resolution single-particle cryo-EM structures of Pma1 and GS as structural references. Pma1, which has a prominent extramembrane domain, could be confidently identified in tomograms. Annotation of GS was more limited due to its relatively small extramembrane domain. In the revised manuscript, we have acknowledged this limitation and focused our analysis of membrane microdomain organization primarily on observations derived from Pma1 distribution. GS distribution was qualitatively assessed using a subset of high-contrast, manually screened tomograms and further supported by fluorescence microscopy of cells expressing YFP-tagged GS (Lines 357–375), which provided a complementary estimate of its spatial distribution at the cellular level.

5) The apparent even dispersion of GS in the cryo-ET reconstructions, regardless of CST treatment, contradicts the manuscript's conclusions. Moreover, the accuracy of GS annotation is questionable due to its small size.

We acknowledge that the punctate GS distribution observed in fluorescence microscopy of cells expressing YFP-tagged GS is not clearly recapitulated in tomograms of crude membranes. This discrepancy likely arises from a combination of technical and biological factors. First, the GS puncta observed by fluorescence microscopy are present only in a subset of cells, appear at relatively low abundance, and are typically on the micron scale. Given the limited field of view and spatial sampling of cryo-ET, the likelihood of capturing such puncta in any single tomogram is inherently low. Second, the higher-order organization of GS may be more dynamic or loosely assembled, which may appear diffuse or less distinct in cryo-ET reconstructions than they are in fluorescence microscopy at the cellular level. In the revised manuscript, we have updated the live-cell imaging section (Lines 366–375) to reflect these caveats and provide a more detailed interpretation of GS spatial organization.

6) The proteomics analysis indicates that Pma1 and GS are significantly less abundant than many other membrane proteins; for example, Pst2 is over 200 times more abundant than Pma1 and GS. However, in the cryo-ET reconstructions, Pma1 and GS appear abundant. This discrepancy is concerning, as it seems improbable to accommodate several hundred times more proteins within the same membranes.

We thank the reviewer for this comment and the opportunity to clarify our analysis. Our mass spectrometry analysis of the crude membrane samples was conducted in two stages. First, we identified a broad set of abundant proteins, including both integral and peripherally associated membrane proteins, as listed in **Supplementary Table 1**. In the second stage, we refined this list to focus specifically on large (>100 kDa) integral membrane proteins, normalized to Fks1 levels, as shown in **Fig. 2A**. This subset includes proteins more likely to be detectable in tomograms of crude membranes.

Many of the highly abundant proteins included in **Supplementary Table 1**, such as Pst2 and Pst3 (quinone oxidoreductases involved in redox homeostasis), are

peripheral membrane proteins known to localize to multiple subcellular compartments, including the cytoplasm and mitochondria. Although they are enriched in the membrane fraction, they are not embedded in the lipid bilayer, and their membrane association is often transient or indirect. Moreover, due to their small size, they fall below the detection limit at our imaging conditions. We have clarified this distinction in the revised manuscript (Lines 135–164).

7) It will be useful to also include proteomics analysis of CST-treated membranes.

As the reviewer suggested, we performed a comparative proteomic analysis of crude membrane fractions from CSF-treated and untreated cells and presented the results in **Supplementary Fig. 9**, which are referenced in the revised manuscript (Lines 376-381).

8) Regarding the live cell fluorescent imaging experiments, additional details are required, particularly concerning CST treatment protocols. It is also essential to quantify YFP-Fks1 expression levels across different experimental conditions.

As recommended by the reviewer, we have added details on CSF treatment conditions in live-cell imaging experiments (Lines 992–995). Additionally, we have included quantitative mass spectrometry data comparing YFP-Fks1 expression levels to those of wild-type Fks1. These results are presented in **Supplementary Fig. 7** in the revised manuscript.

Reviewer #2: I co-reviewed this manuscript with one of the reviewers who provided the listed reports. This is part of the Nature Communications initiative to facilitate training in peer review and to provide appropriate recognition for Early Career Researchers who co-review manuscripts.

Thank you for reviewing our manuscript.

Reviewer #3:

Comment:

MP proteins are considered potential targets for new antifungal treatments. This is also the rationale for the current study.

PMA1 proteins are found to form the rosette like structures (hexamers) as seen before in cryo structures of PMA1. (Heit et al. Structure of the hexameric fungal plasma membrane proton pump in its autoinhibited state. (2021) Sci Adv. doi: 10.1126/sciadv.abj5255.); Zhao et al (2021). Structure and activation mechanism of the hexameric plasma membrane H⁺-ATPase. Nat Commun. doi: 10.1038/s41467-021-26782-y.).

In the present work, PMA1 proteins are found in subdomains appearing to induce curvature, it is however not demonstrated if this of biological importance.

We thank the reviewer for highlighting this observation. Single-particle cryo-EM studies by Heit et al. (2021) and Zhao et al. (2021) revealed that hexameric Pma1 complexes contain crystalline-packed lipid molecules within their central cavity. These lipids, likely ordered sphingolipids, were proposed to stabilize the hexameric conformation.

In our study, we observed that Pma1-enriched microdomains frequently coincide with localized membrane depressions. This curvature change may reflect a higher local composition of ordered sphingolipids relative to adjacent regions, consistent with the presence of tightly packed integral lipids in the Pma1 cryo-EM structures. Although the biological significance of this curvature remains to be fully understood, our findings support a potential role of integral lipids in higher-order organization of Pma1 on the membrane. This interpretation has been incorporated into the revised manuscript (Lines 186–192).

The workflow described in the paper is of great value to the scientific field and might have applications in developing new drugs in combination with other studies.

How this paper might help lead to the direct biological understanding the mode of action of echinocandins might be further away, and the biological question answered seems a bit far-fetched. The work documents very well the organization of the plasma membrane in response to caspofungin treatment, but the implication for antifungal action remains poorly understood.

We don't find it surprising that the membrane surface tension is affected when treating the spheroblasts with CSF, as CSF has detergent-like properties (hydrophilic hexapeptide headgroup and hydrophobic aliphatic tailgroup) and as spheroblasts generally are known to be very fragile. What is the effect of adding a simpler amphiphilic molecule? We suggest including a control with a relevant detergent that will integrate in the membrane but not specifically target GS.

We thank the reviewer for this insightful comment. We concur that the amphipathic nature of echinocandin compounds bears a resemblance to detergents. It is indeed striking that, despite this characteristic, echinocandins exhibit such high specificity toward GS in both clinical and experimental settings. We agree that elucidating the basis of this specificity is an important area of investigation and may ultimately yield valuable insights into the molecular mechanisms governing echinocandin activity.

In the past two years, high-resolution single particle cryoEM studies of GS have provided important insights into this perspective and demonstrated that detergents can indeed alter GS conformation and its interaction with echinocandin drugs.

In the study by Hu et al. (Nature, 2023; DOI: 10.1038/s41586-023-05856-5), GS purified from *S. cerevisiae* using different detergents exhibited divergent responses to echinocandin treatment. Notably, GS purified with GDN displayed increased enzymatic activity upon exposure to CSF, rather than inhibition. These findings suggest that

detergent may displace or disrupt integral lipids critical for GS function, thereby altering its conformation, activity, and drug susceptibility.

Similarly, Zhao et al. (Science Advances, 2023; DOI: 10.1126/sciadv.adh7820) reported a GS structure purified with a different detergent. Although the overall GS architecture was preserved (RMSD ~1.7 Å) compared to the structure by Hu et al., significant structural differences were observed near the active site, and the integral lipid densities within GS structures varied between the two preparations.

Together, these studies revealed that detergent can perturb GS structure and function. At the same time, they highlight the essential role of integral lipids in maintaining GS structural and functional integrity.

Based on these findings, we would expect that introducing an exogenous detergent in cells or spheroplasts would likely compromise membrane integrity and introduce substantial perturbations to the structure and function of GS. However, treatment with detergent may perturb other membrane-associated complexes. This would complicate the interpretation of any observed effects. While such an experiment might provide some insight into echinocandin drug action, it would be challenging to distinguish specific effects on GS from broader biophysical disruptions to the membrane environment.

The authors use a mutant yeast strain to show that certain lipids are important for CSF activity (mainly sphingolipids). However, we lack some functional data in this study to understand the effect of CSF on the two proteins. We suggest including functional data on PMA1 and GS to test the effect of CSF treatment.

We thank the reviewer for this valuable suggestion. In the original manuscript, we included drug inhibition assays comparing the minimal inhibitory concentration (MIC) values of the Fks1-overexpression and Δ fen1 strains to those of the wild-type controls (**Supplementary Fig. 1**). These results support the conclusion that elevated GS expression reduces susceptibility to CSF, while disruption of sphingolipid biosynthesis alters the cellular response to CSF treatment.

To further examine the impact of CSF treatment on GS function, we expanded our analysis in the revised manuscript to include additional drug interaction studies targeting sphingolipid biosynthesis. Specifically, we combined CSF treatment with either myriocin (an inhibitor of serine palmitoyltransferase) or phytosphingosine (PHS) supplementation to modulate sphingolipid biosynthesis (new **Fig. 6**, Lines 440–460). In both cases, no significant pharmacological interactions with CSF were detected in wild-type cells, although it seems that inhibition of long-chain base synthesis by myriocin partially recovers echinocandin susceptibility in the Δ fen1 mutant. Since myriocin inhibits LCB synthesis, it may help rebalance the VLCFA-to-LCB ratio toward wild-type levels, thereby altering the membrane lipid environment surrounding GS complexes and enhancing their sensitivity to CSF.

These pharmacological approaches complement our analysis of the Δ fen1 mutant and provide deeper insight into how specific alterations in sphingolipid biosynthesis influence CSF activity at the cellular level.

We also evaluated the effect of fluconazole (FLC), an inhibitor of ergosterol synthesis, in combination with CSF or micafungin (MCF) in wild-type and Δ fen1 strains (new **Supplementary Fig. 11**, Lines 470-481). Results showed that there were no synergistic interactions observed between FLC and either CSF or MCF in the wild type strain and a mild synergistic interaction in the Δ fen1 mutant strain. This suggests that ergosterol does not significantly influence GS binding or echinocandin susceptibility unless sphingolipid biosynthesis is perturbed.

Regarding Pma1, echinocandins are highly specific for GS and do not directly target Pma1. However, to assess whether CSF treatment exerts any indirect effects on Pma1 expression or abundance, we conducted quantitative proteomic analyses of crude membrane preparations across all experimental strains. These data (new **Supplementary Figs. 7 and 9**) show that Pma1 levels at the plasma membrane exhibit only a slight increase following CSF treatment, likely due to the loss of loosely associated membrane proteins, and remain largely unchanged in the Δ fen1 background. A modest reduction in Pma1 abundance was observed in the Fks1-overexpression strain, possibly due to spatial crowding. Collectively, these findings suggest that the large-scale reorganization of Pma1 observed in our study is unlikely to result from altered protein abundance and more likely reflects changes in the membrane's biophysical properties.

Quality of work

The experiments seem to be performed thoroughly with an appropriate number of controls and repetitions.

The manuscript is well documented with a comprehensive supplementary data.

The manuscript is well written in a concise manner.

Reviewer #4:

The paper by Jiang et al. describes the perturbation of fungal plasma membrane upon treatment with the antifungal drug caspofungin. The authors found that caspofungin perturbs the distribution of the polysaccharide beta-glucans and the plasma membrane proton pump Pma1. Overall, the methodology used is the first of its kind. The results are well organized, well-presented and well-discussed. Few points are below.

1) Fen1 is an enzyme for the synthesis of very long chain fatty acids, which are fatty acids used to make various lipids, including sphingolipids. Thus, a mutant lacking Fen1 will have a defect in the synthesis of lipids, including sphingolipids, containing very long chain fatty acids. Thus, it is possible that the defects observed in the paper by the authors in the delta-fen1 may be due to the lack of the lipids other than sphingolipids. Please clarify these issues.

Indeed, sphingolipid biosynthesis is a complex pathway in which very long-chain fatty acids (VLCFAs) are conjugated with long-chain bases (LCBs), such as dihydrosphingosine (DHS) and phytosphingosine (PHS), to produce complex sphingolipids. Fen1 encodes a fatty acid elongase that functions in the early steps of

VLCFA synthesis. Deletion of *FEN1* reduces VLCFA levels and may also impact other lipid classes within the pathway.

To further investigate whether/how the reduced susceptibility to CSF observed in the $\Delta fen1$ strain is linked to alterations in VLCFA and LCB levels, we included additional drug interaction assays in the revised manuscript. Specifically, we modulated intracellular LCB levels using myriocin, an inhibitor of serine palmitoyltransferase (which blocks LCB synthesis), and phytosphingosine (PHS) supplementation (to increase LCB levels). These checkerboard assays (new **Fig. 6**, Lines 440–460) were designed to test whether perturbing LCB levels affects CSF susceptibility, independent of reduced VLCFA levels. The results suggest that while elevated LCB levels may partially contribute to the decreased CSF susceptibility in the $\Delta fen1$ mutant, increasing LCB levels alone was insufficient to alter GS-CSF interactions or drug susceptibility.

2) Have the authors considered using different mutants more specifically involved in the sphingolipid (or ergosterol) pathways? Alternatively, drugs that inhibit specifically enzymes involved in the synthesis of sphingolipids exist. Have the authors considered using them in combination with caspofungin?

Previous studies have shown that *C. glabrata* mutants lacking genes involved in sphingolipid biosynthesis and lipid transport, such as *FEN1* (fatty acid elongase), *CKA2* (casein kinase II), and *LEM3* (a phospholipid flippase), exhibit morphological abnormalities and altered susceptibility to echinocandins. Among these, we selected the $\Delta fen1$ mutant for detailed analysis because *FEN1* encodes a key enzyme involved in the early steps of very long-chain fatty acid (VLCFA) synthesis, a crucial branch of the sphingolipid pathway. This model allowed us to systematically investigate how targeted disruption of sphingolipid biosynthesis modulates CSF activity.

We did not pursue genetic mutants in the ergosterol biosynthesis pathway, as checkerboard assays combining fluconazole (FLC), an inhibitor of ergosterol synthesis, with CSF revealed no significant interaction or evidence that ergosterol modulates CSF activity in the wild type cells (new **Supplementary Fig. 11**, Lines 470-481).

In addition to the genetic approach, and as suggested by the reviewer, we also implemented a pharmacological strategy to more specifically perturb distinct steps in sphingolipid biosynthesis. In the revised manuscript, we include drug interaction studies using myriocin, an inhibitor of serine palmitoyltransferase, and PHS supplementation to modulate long-chain base (LCB) levels. These experiments allowed us to further assess how specific changes in sphingolipid composition and abundance influence CSF susceptibility. The results are presented in **Fig. 6** and discussed in the Results section (Lines 440–460).

3) Plasma membrane perturbations are clearly visible upon caspofungin treatment. But ergosterol is a key component of the fungal plasma membrane. Have the authors considered using in their assay an antifungal diminishing ergosterol synthesis (e.g. fluconazole) or altering ergosterol function (e.g. amphotericin b)? These drugs may or may not cause perturbations in the plasma membrane similar to those caused by

caspofungin (e.g. beta-glucan/Pma1).

As noted above, we now included Fluconazole (FLC) treatment, targeting ergosterol biosynthesis, tested in combination with CSF and micafungin in both wild-type and Δ fen1 strains, to evaluate the contribution of ergosterol to echinocandin susceptibility (see **Supplementary Fig. 11**, Lines 470-481). Results showed that there were no synergistic interactions observed between FLC and either CSF or MCF in the wild type strain and a mild synergistic interaction in the Δ fen1 mutant strain. This suggests that ergosterol does not significantly influence GS binding or echinocandin susceptibility unless sphingolipid biosynthesis is perturbed.

Reviewer #5: I co-reviewed this manuscript with one of the reviewers who provided the listed reports. This is part of the Nature Communications initiative to facilitate training in peer review and to provide appropriate recognition for Early Career Researchers who co-review manuscripts.

Thank you for reviewing our manuscript.